# CHC22 clathrin recruitment to the early secretory pathway requires two-site interaction with SNX5 and p115

Joshua Greig [iD] [1,2,4], George T Bates [iD] [1,2,4], Daowen I Yin [iD] [1,2], Kit Briant [iD] [1,2], Boris Simonetti[3], Peter J Cullen [iD] [3] & Frances M Brodsky [iD] [1,2 ✉]

## Abstract

**The two clathrin isoforms, CHC17 and CHC22, mediate separate intracellular transport routes. CHC17 performs endocytosis and housekeeping membrane traffic in all cells. CHC22, expressed most highly in skeletal muscle, shuttles the glucose transporter GLUT4 from the ERGIC (endoplasmic-reticulum-to-Golgi intermediate compartment) directly to an intracellular GLUT4 storage compartment (GSC), from where GLUT4 can be mobilized to the plasma membrane by insulin. Here, molecular determinants distinguishing CHC22 from CHC17 trafficking are defined. We show that the C-terminal trimerization domain of CHC22 interacts with SNX5, which also binds the ERGIC tether p115. SNX5, and the functionally redundant SNX6, are required for CHC22 localization independently of their participation in the endosomal ESCPE-1 complex. In tandem, an isoform-specific patch in the CHC22 N-terminal domain separately mediates binding to p115. This dual mode of clathrin recruitment, involving interactions at both N- and C-termini of the heavy chain, is required for CHC22 targeting to ERGIC membranes to mediate the Golgi-bypass route for GLUT4 trafficking. Interference with either interaction inhibits GLUT4 targeting to the GSC, defining a bipartite mechanism regulating a key pathway in human glucose metabolism.**

**Keywords** Clathrin; CHC22; Sorting nexin 5; Golgi bypass; p115
**Subject Category** Membranes & Trafficking

## Introduction

Clathrin coats mediate some of the major routes for vesicular intracellular transport (Briant et al, 2020). Humans, along with several other vertebrate species, express two types of clathrin, each comprising a separate isoform of clathrin heavy chain, CHC17 and CHC22, named for their encoding human chromosomes

(Fumagalli et al, 2019; Wakeham et al, 2005). CHC17 clathrin is the canonical form that performs receptor-mediated endocytosis and a range of intracellular trafficking pathways in all eukaryotic tissues (Briant et al, 2020). CHC22 clathrin, is expressed most highly in skeletal muscle, and is essential for trafficking the insulin-responsive glucose transporter GLUT4 to an intracellular storage compartment (GSC) where GLUT4 awaits insulin-induced translocation to the cell surface (Camus et al, 2020; Esk et al, 2010; Hoshino et al, 2013; Vassilopoulos et al, 2009). This translocation pathway is responsible for 70% of post-prandial glucose clearance from the bloodstream in humans (Bogan, 2012; Gould et al, 2020; Leto and Saltiel, 2012) and malfunction of GLUT4 membrane traffic is associated with Type 2 diabetes (Bogan, 2012). CHC22 shares 85% sequence identity with CHC17, yet is distinct in both its intracellular recruitment profile and biochemical properties (Dannhauser et al, 2017). For GLUT4 targeting to the GSC, CHC22 operates at the Endoplasmic Reticulum to Golgi Intermediate Compartment (ERGIC), bypassing the Golgi, in a pathway that is not mediated by CHC17 (Camus et al, 2020). This study addresses the molecular mechanism of this CHC22 recruitment to the early secretory pathway to establish how its membrane association is regulated and ultimately how CHC22 might be manipulated to affect human glucose metabolism.

The main mechanism by which CHC17 is recruited to intracellular membranes is via the binding of its N-terminal domain (TD) to a variety of adaptors that recognize both phosphoinositides and membrane-embedded cargo. These include the heterotetrameric adaptors AP1 and AP2, which recruit CHC17 to endosomes and the plasma membrane respectively, as well as the monomeric GGA adaptors that enable protein sorting at endosomes and the *trans*-Golgi network (Briant et al, 2020; Lemmon and Traub, 2012; Robinson and Bonifacino, 2001). The CHC17 TD has a seven-bladed β-propeller structure that binds adaptors at four identified sites (Lemmon and Traub, 2012). Equivalent binding sites on the CHC22 TD are predicted by sequence homology. However, CHC22 does not bind AP2, nor functions at the plasma membrane in receptor-mediated endocytosis, while demonstrating preferential binding to GGA2 compared to CHC17 (Dannhauser et al, 2017; Liu et al, 2001). This latter interaction, plus CHC22

[1]Structural and Molecular Biology, Division of Biosciences, University College London, London WC1E 6BT, UK. [2]Institute of Structural and Molecular Biology, Birkbeck and University College London, London WC1E 7HX, UK. [3]School of Biochemistry, Faculty of Life Sciences, University of Bristol, Bristol, UK. [4]These authors contributed equally: Joshua Greig, George T Bates. ✉E-mail: f.brodsky@ucl.ac.uk

binding to AP1, may account for CHC22 function in sorting GLUT4 from endosomes back to the GSC following re-uptake from the plasma membrane by CHC17 after insulin-stimulated GLUT4 release (Esk et al, 2010; Gould et al, 2020). CHC22 function at the ERGIC to form the GSC is attributed to its isoform-specific participation in a complex that includes the ERGIC tether protein p115 and insulin-responsive-amino peptidase (IRAP), which binds both p115 and GLUT4 (Camus et al, 2020; Hosaka et al, 2005), but whether this involves binding by the CHC22 TD and/or requires additional partner proteins, has not been established.

Yeast-two-hybrid assays identified sorting nexin 5 (SNX5) as a protein that binds CHC22 but not CHC17 (Towler et al, 2004) in a domain that for CHC17 interacts with clathrin light chain (CLC) subunits (Chen et al, 2002; Wilbur et al, 2010). While yeast two-hybrid assays could also detect CLC binding to this domain of CHC22, the CLCs do not appear to associate with CHC22 in cells (Dannhauser et al, 2017; Liu et al, 2001). SNX5 and the functionally-redundant paralogue protein SNX6 comprise a membrane-remodeling BAR domain and a Phox homology (PX) domain, with the latter binding cargo instead of the canonical lipid PX domain ligands (Kvainickas et al, 2017; Simonetti et al, 2019). Although initially implicated in Retromer sorting at endosomes by homology with yeast proteins (Rojas et al, 2007; Simonetti and Cullen, 2018; van Weering et al, 2012; Wassmer et al, 2007), it is now established that SNX5/6 are subunits of the ESCPE-1 complex, which mediates endosomal sorting distinct from Retromer pathways (Evans et al, 2020; Simonetti et al, 2022; Simonetti et al, 2019). In ESCPE-1, SNX5/6 heterodimerize with SNX1, or its functionally-redundant paralogue protein SNX2, to interact with lipids through the PX domain of SNX1/2 and with cargo through the PX domain of SNX5/6 (Evans et al, 2020; Kvainickas et al, 2017; Simonetti et al, 2017; van Weering et al, 2012).

Here, we delineate the relationship between the differential membrane recruitment of CHC22 relative to CHC17, and the unique binding partners identified for CHC22. We confirm that CHC22 binds endogenously to SNX5/6 in cells and demonstrate that this interaction is exclusive of SNX5/6 participation in the ESCPE-1 complex. We show that SNX5 directly binds p115 and that this interaction is required but not sufficient for CHC22 localization and for GSC formation. The CHC22-ERGIC interaction and GSC formation were found to additionally require binding of the CHC22 TD directly to p115. This binding is disrupted by the mutation of a divergent patch in the CHC22 TD to the equivalent residues in CHC17. We thus demonstrate that the specific localization of CHC22 to early secretory membranes requires a bipartite recruitment mechanism. CHC22 is targeted to these membranes by recognition of p115 directly via the CHC22 N-terminal domain, and indirectly through SNX5/6 bridging at the C-terminal trimerization domain. The demonstration that both interactions are essential for the formation of an insulin-responsive GSC defines key molecular determinants regulating glucose clearance in humans with potential for therapeutic targeting.

# Results

## SNX5/6 are required for correct localization of CHC22

CHC22 is present in both HeLa and muscle cells and functions at the ERGIC for GLUT4 transport to the GSC in a pathway distinct from CHC17 clathrin traffic (Camus et al, 2020). In HeLa cells, which endogenously express CHC22 at levels comparable to myoblasts (Esk et al, 2010), CHC22 localizes to a perinuclear region overlapping strongly with p115 (Camus et al, 2020). To identify determinants of the perinuclear localization of CHC22, a quantitative image analysis methodology was developed. In brief, DAPI-stained nuclei were used to identify each cell, as well as provide a positional marker for quantification of the perinuclear area. The area between the nuclear boundary and a dilated boundary extending a further 4.5 μm created a region comprising p115 staining and ERGIC structures, in which the perinuclear mean fluorescence intensity (PeriMI) of CHC22 was measured (Fig. 1A).

To assess the effect on CHC22 distribution when SNX5 levels were altered, a HeLa cell line engineered by CRISPR to delete the genes encoding SNX5 and SNX6 (ΔSNX5/6) was analyzed (Fig. 1B). This SNX5/6-null cell line was previously produced for and characterized in studies of Retromer and ESCPE-1 trafficking (Simonetti et al, 2022; Simonetti et al, 2017; Simonetti et al, 2019), and the deletion was confirmed in this study (Fig. EV1A). In previous studies, phenotypes were only observed when both SNX5 and SNX6 were deleted, supporting their functional redundancy. Compared to the control parental HeLa line from which they were derived (Con), CHC22 in ΔSNX5/6 cells exhibited reduced perinuclear enrichment (Fig. 1B) and quantification of the CHC22 PeriMI (Fig. 1C) revealed a significant reduction ($P = 0.0095$) in the ΔSNX5/6 cell line. This reduced perinuclear enrichment was verified by measuring the association of CHC22 with its ERGIC interactor p115 using a Proximity Ligation Assay (PLA), where fluorescence puncta represent labelling of CHC22 and p115 molecules within 40 nm of each other (Alam, 2018) (Fig. 1D). Quantification of the average number of puncta per cell confirmed that fewer PLA puncta were observed in ΔSNX5/6 cells compared to parental control HeLa cells ($P = 0.03$, Fig. 1E), supporting a role for SNX5/6 in CHC22 enrichment to p115-containing membranes. Moreover, a reduction of CHC22 membrane association was observed following fractionation of the ΔSNX5/6 cells and quantifying the percentage of total CHC22 associated with membranes by immunoblotting (Fig. 1F,G). The relative percentage of CHC22 in the membrane fraction was decreased in ΔSNX5/6 cells compared to parental control cells ($P = 0.0002$, Fig. 1G), while the proportion of CHC17 associated with membranes did not significantly differ between the control (C) and null cell (Δ) lines ($P = 0.94$, Fig. 1G). Consistent with previous reports, a higher proportion of CHC17 was present in the cytosolic fraction and a greater proportion of CHC22 on membranes (Liu et al, 2001) in all cells. Moreover, the distribution of the cellular fractionation marker HSC70 was indistinguishable between the control and null cell line ($P = 0.49$, Fig. 1G), further indicating a specific effect on CHC22 membrane association in the SNX5/6 null cells.

An active role for SNX5 in CHC22 membrane localization was then investigated to rule out indirect effects of SNX5/6 loss. When control HeLa cells were transfected to overexpress FLAG-tagged recombinant SNX5 (Towler et al, 2004) (Fig. 1H), the PeriMI of CHC22 increased by a magnitude of approximately three-fold over mock transfected cells ($P < 0.0001$, Fig. 1I). Additionally, transfection of the ΔSNX5/6 cells with FLAG-tagged SNX5 restored perinuclear localization of CHC22 (Fig. 1J), while mock transfected ΔSNX5/6 cells lacked perinuclear CHC22. Quantification showed increased CHC22 PeriMI in the rescued cells relative to control ($P < 0.0001$, Fig. 1K), supporting a role for SNX5 in the perinuclear

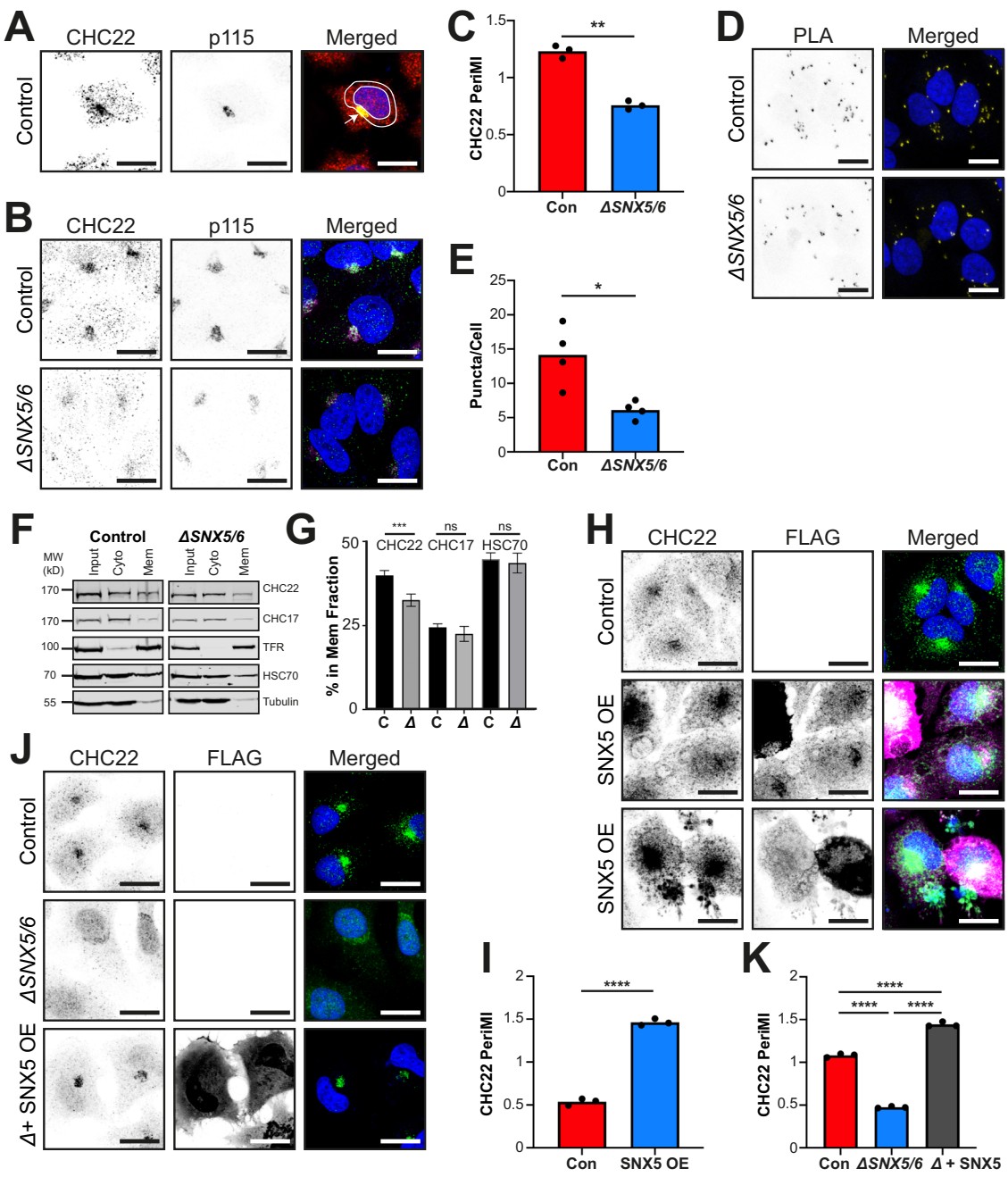

localization of CHC22 and demonstrating that SNX5 alone can compensate for the loss of both SNX5 and SNX6. Together, these data indicate that CHC22 perinuclear enrichment positively correlates with the levels of SNX5/6 present in cells.

## SNX5/6 role in CHC22 localization is independent of ESCPE-1 and Retromer

The requirement of SNX5/6 for CHC22 membrane localization raises the question of whether known partners of SNX5/6 are also involved. To address this, CHC22 localization was tested in HeLa cells lacking SNX1/2, which dimerize with SNX5/6 in the endosomal ESCPE-1 complex, and in cells lacking the Retromer complex (VPS26A/B:VPS35:VPS29) formerly considered a SNX5/6 interactor (Simonetti et al, 2022; Simonetti et al, 2017; Simonetti et al, 2019). The retromer-null (ΔVPS35) and SNX1/2-null cells (ΔSNX1/2) were produced by CRISPR engineering and characterized in earlier studies (Simonetti et al, 2022; Simonetti et al, 2017; Simonetti et al, 2019) with deletion confirmed in this study (Fig. EV1B). In the ΔVPS35 cells, neither CHC22 ($P = 0.053$, Fig. 2A,B) nor p115 ($P = 0.673$, Fig. 2A,C) perinuclear intensities were altered relative to the control, while a decrease was observed in the PeriMI of the cis-Golgi marker GM130 ($P < 0.0001$, Fig. 2A,D), a predicted indirect effect from disruption of endosomal

**Figure 1.   SNX5/6 levels correlate with localization of CHC22 to perinuclear membranes.**

(A) Representative images of control HeLa cells immunolabeled for CHC22 (red in merged), and p115 (green in merged) with single channels in black and white. Merged image shows all channels with overlap in yellow and includes DAPI-stained nuclei (blue). Arrow indicates the compartment overlap in the perinuclear region. Two concentric white circles mark the boundary of the perinuclear area measured using the perinuclear mean intensity (PeriMI) method. (B) Representative images of control parental HeLa cell line (top) and CRISPR-mediated SNX5/6 null (*ΔSNX5/6*) HeLa-derived line (bottom) immunolabeled for CHC22 (green in merged) and p115 (magenta in merged), nuclei stained with DAPI (blue in merged). (C) Quantification of PeriMI for CHC22 in the cell lines shown in (B). Each bar represents the mean normalized PeriMI value for three independent experiments represented individually as dots (10–20 cells per genotype, per experiment). Statistical analysis was performed using a two-tailed Student's *t* test with Welch's correction (*P* = 0.0095). (D) Representative images of control parental HeLa line (top) and *ΔSNX5/6* HeLa-derived line (bottom), with CHC22 and p115 targeting antibodies labelled using Proximity Ligation Assay (PLA) probes. PLA-positive puncta are shown in black and white in single channel images and in yellow in merged, with DAPI-stained nuclei (blue). (E) Quantification of the CHC22-p115 PLA puncta shown in (D) measuring the average number of puncta per cell between the genotypes. Each bar represents the median for each genotype from four independent experiments, with each repeat individually depicted as a dot (23–51 cells per genotype, per experiment). Statistical analysis was performed using a two-tailed Student's *t* test with Welch's correction (*P* = 0.0307). (F) Representative immunoblot for membrane fractionation of lysate from control HeLa cells (left) and *ΔSNX5/6* HeLa cells (right) immunoblotted for CHC22, CHC17, transferrin receptor (TfR), HSC70, and tubulin. Lanes show the input, cytosolic (Cyto) fraction, and membrane (Mem) fraction respectively. The migration positions of molecular weight (MW) markers are indicated at the left in kilodaltons (kD). (G) Quantification of the percentage of CHC22, CHC17, and HSC70 protein associated with cellular membranes ([Mem/(Mem +Cyto)]x100) for control (C) and *ΔSNX5/6* (Δ) cells. Graphs show mean ± SEM. Statistical analysis was performed using a two-tailed Student's *t* test with Welch's correction (*P* = 0.0002, *P* = 0.94, *P* = 0.49 from left to right, *n* = 3 independent experiments). (H) Representative images of control HeLa cells either mock-transfected (top) or transfected to overexpress FLAG-SNX5 (SNX5 OE) (middle and bottom). Two rows are shown to demonstrate the range of transfection effects. Cells were immunolabeled for CHC22 (green in merged) and FLAG (magenta in merged) with single channels shown in black and white and nuclei stained with DAPI (blue in merged). (I) Quantification of CHC22 PeriMI for mock (Con)- and FLAG-SNX5 (OE)-transfected HeLa cells. Each bar represents the mean normalized PeriMI value from three independent experiments, each individually depicted as dots (14–25 cells per condition, per experiment). Statistical analysis was performed using a two-tailed Student's *t* test with Welch's correction (*P* < 0.0001). (J) Representative images of mock-transfected HeLa cells (top), mock-transfected *ΔSNX5/6* HeLa cells (middle), and *ΔSNX5/6* HeLa cells transfected to overexpress FLAG-SNX5 (*ΔSNX5/6* + SNX5 OE) (bottom). Cells were immunolabeled for CHC22 (green in merged) and FLAG (not shown in merged, for clarity) with single channels shown in black and white and nuclei stained with DAPI (blue in merged). (K) Quantification of CHC22 PeriMI for HeLa mock-transfected cells (Con), *ΔSNX5/6* HeLa mock-transfected cells, and *ΔSNX5/6* HeLa cells transfected to overexpress FLAG-SNX5 (Δ + SNX5 OE) shown in (J). Each bar represents the mean normalized PeriMI value from three independent experiments, each individually depicted as dots (8–17 cells per genotype, per experiment). Statistical analysis was performed using a one-way ANOVA with a Tukey post-hoc test (*P* < 0.0001 for all three). ns = not significant; *\*P* < 0.05; *\*\*P* < 0.01; *\*\*\*P* < 0.001; *\*\*\*\*P* < 0.0001. Scale bars: 25 μm. Source data are available online for this figure.

traffic compartments (Wassmer et al, 2007). By contrast, in *ΔSNX5/6* cells (Fig. 2A–E), CHC22 PeriMI was reduced (*P* < 0.0001, Fig. 2B), as already observed (Fig. 1C–E) and the mean intensity of p115 was also reduced (*P* = 0.0004, Fig. 2C), with no change detected in GM130 relative to control (*P* = 0.11, Fig. 2D). A decrease in the CHC22 total protein level in *ΔSNX5/6* cells was observed (Fig. 2E,J), suggesting that the presence of SNX5/6 stabilizes CHC22, which in turn, stabilizes perinuclear enrichment of p115. Together, these data indicate that the Retromer pathway is distinct from the CHC22-SNX5/6 pathway and suggest a role for SNX5/6 in linking CHC22 with p115.

By comparison, an increase in CHC22 PeriMI (*P* < 0.0001, Fig. 2F, G) was observed in *ΔSNX1/2* cells relative to control cells with no change in p115 PeriMI (*P* = 0.15, Fig. 2H) and an increase in GM130 PeriMI (*P* = 0.0006, Fig. 2I). Again, the PeriMI of CHC22 was reduced in *ΔSNX5/6* cells in parallel with effects on p115 (Fig. 2F–H). Levels of CHC22 protein were increased in the *ΔSNX1/2* cells (Fig. 2J), a stability phenotype that might be expected if increased SNX5/6 is available to interact with CHC22 when not participating in the ESCPE-1 complex with SNX1/2. This could also potentially account for the increased CHC22 perinuclear localization in these cells, similar to the effect observed following SNX5 overexpression (Fig. 1H). To assess whether the perinuclear localization of SNX5/6 increases in the absence of SNX1/2, *ΔSNX1/2* cells were immunolabeled for SNX6, since no available antibodies that recognize SNX5 were found to function for immunofluorescent staining (Fig. 2K). Quantification showed a significant increase of SNX6 PeriMI in *ΔSNX1/2* cells (*P* = 0.0067, Fig. 2L). The enhanced CHC22 PeriMI in *ΔSNX1/2* cells is consistent with dependence on SNX5/6 for its localization and with independence from participation of the ESCPE-1 dimer in its recruitment.

The decreased PeriMI of GM130 in *ΔVPS35* cells (Fig. 2A,D) and its increased PeriMI in *ΔSNX1/2* cells (Fig. 2F,I) did not correspond with altered CHC22 localization in these cells, and further emphasized the distinction between the CHC22 localization and sorting endosome traffic affecting GM130. Likewise, localization of transferrin receptor (TfR), the classical early endocytic cargo (Mayle et al, 2012), in the *ΔSNX1/2*, *ΔVPS35* and *ΔSNX5/6* cells showed expected endosomal effects distinct from changes in CHC22 distribution in these cells (Fig. EV1C,D). TfR PeriMI was elevated in the *ΔSNX1/2* and *ΔVPS35* cells, and also by a small but significant level in the *ΔSNX5/6* cells, consistent with loss of endosomal recycling to the cell surface and previous analysis of TfR distribution when Retromer components are depleted (Chen et al, 2013; Tabuchi et al, 2010). These observations further support the functional distinction between SNX5/6 and Retromer (Simonetti et al, 2017; Simonetti et al, 2019), as well as the functional segregation of SNX5/6 from SNX1/2, as seen for CHC22 localization. Overall, these data indicate that the CHC22-SNX5/6 interaction is distinct from both the Retromer pathway and the ESCPE-1 complex.

## SNX5 binding to the CHC22 trimerization domain (TxD) is exclusive of binding to SNX1

An earlier study, supporting yeast-two-hybrid evidence of CHC22-SNX5 interaction, demonstrated that FLAG-tagged SNX5 overexpressed in cells could be co-immunoprecipitated with CHC22 (Towler et al, 2004). To establish whether this interaction occurs between endogenous proteins at physiological expression levels, CHC22 was immunoprecipitated from HeLa cell lysate and from lysate of human muscle myotubes differentiated from AB1190 myoblasts (Camus et al, 2020), and the co-immunoprecipitated proteins analyzed by immunoblotting. SNX5 and SNX6 co-immunoprecipitated with CHC22 from both cell lysate

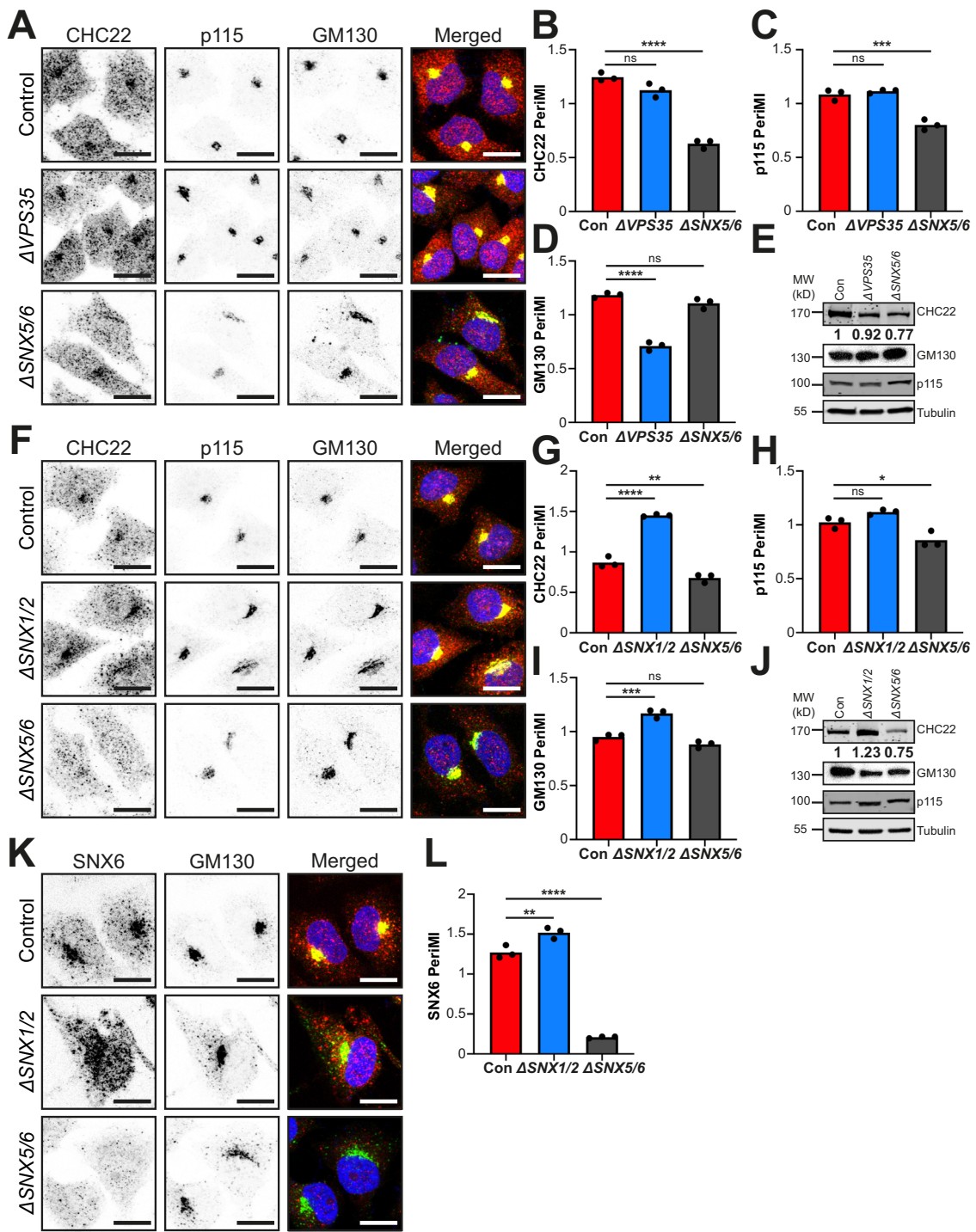

preparations (Fig. 3A,B). Thus, the CHC22-SNX5/6 interaction is relevant to CHC22 function in muscle (Camus et al, 2020), as well as in HeLa cells. Further examination of the CHC22 co-immunoprecipitates revealed that neither SNX1 nor the Retromer component VPS35 were present in complex with CHC22 (Fig. EV2), additionally supporting the lack of involvement of the Retromer and ESCPE-1 complexes.

Previous yeast-two-hybrid studies mapped the CHC22-SNX5 interaction to the Hub domain of CHC22 (C-terminal third, residues 1074–1640, Fig. 3C) and to residues 238–404 of the SNX5 BAR domain and detected no interaction between SNX5 and the CHC17 Hub domain (Towler et al, 2004). To model the configuration and positioning of SNX5-CHC22 interaction, the partner protein sequences were simulated using AlphaFold

**Figure 2.**    SNX5/6-dependent CHC22 localization is independent of both VPS35 retromer and SNX1/2 components of the ESCPE-1 complex.

(A) Representative images of control parental HeLa cell line (top), VPS35-null (ΔVPS35) HeLa-derived line (middle), and ΔSNX5/6 HeLa-derived line (bottom) immunolabeled for CHC22 (red in merged), p115 (magenta in merged) and GM130 (green in merged) with overlap of labels (yellow) and DAPI-stained nuclei (blue) in merged. (B–D) Quantification of PeriMI for CHC22 (B) ($P = 0.0534$ (ns), $P < 0.0001$(****)), p115 (C) ($P = 0.6734$ (ns), $P = 0.0004$ (****)), and GM130 (D) ($P < 0.0001$ (****), $P = 0.1133$ (ns)), in control, ΔVPS35, and ΔSNX5/6 HeLa cells. Each bar represents the mean normalized PeriMI value from three independent experiments, each individually depicted as dots (18–29 cells per genotype, per experiment). Statistical analysis was performed using a one-way ANOVA with a Tukey post-hoc test. (E) Representative immunoblot of lysates from control (con), ΔVPS35 and ΔSNX5/6 HeLa lines immunoblotted for CHC22, GM130, p115, and tubulin. Normalized mean optical density (OD) values for CHC22 from 3 experiments are shown. The migration position of MW markers is indicated at the left in kilodaltons (kD). (F) Representative images of control parental HeLa cell line (top), ΔSNX1/2 HeLa-derived line (middle), and ΔSNX5/6 HeLa-derived line (bottom) immunolabeled for CHC22 (red in merged), p115 (magenta in merged) and GM130 (green in merged) with overlap of labels (yellow) and DAPI-stained nuclei (blue) in merged. (G–I) Quantification of PeriMI for CHC22 (G) ($P < 0.0001$(****), $P = 0.0085$ (**)), p115 (H) ($P = 0.1519$ (ns), $P = 0.0231$ (*)), and GM130 (I) ($P = 0.0006$ (***), $P = 0.1094$ (ns)), in control, ΔSNX1/2, and ΔSNX5/6 HeLa cells. Each bar represents the mean normalized PeriMI value from three experiments, each individually depicted as dots (15–26 cells per genotype, per experiment). Statistical analysis was performed using a one-way ANOVA with a Tukey post-hoc test. (J) Representative immunoblot of lysates from parental HeLa control cells (con), ΔSNX1/2 cells and ΔSNX5/6 cells immunoblotted for CHC22, GM130, p115, and tubulin. Normalized mean OD values for CHC22 from 3 experiments are shown. The migration positions of MW markers are indicated at the left in kD. (K) Representative images of control parental HeLa line (top), ΔSNX1/2 HeLa-derived line (middle) and ΔSNX5/6 HeLa-derived line (bottom) immunolabeled for SNX6 (red in merged) and GM130 (green in merged) with overlap of labels (yellow) and DAPI-stained nuclei (blue) in merged. (L) Quantification of PeriMI for SNX6 in control, ΔSNX1/2, and ΔSNX5/6 HeLa cells. Each bar represents the mean normalized PeriMI value from three independent experiments, each individually depicted as dots (16–27 cells per genotype, per experiment). Statistical analysis was performed using a one-way ANOVA with a Tukey post-hoc test ($P = 0.0067$ (**), $P < 0.0001$ (****)). ns = not significant; *$P < 0.05$; **$P < 0.01$; ***$P < 0.001$; ****$P < 0.0001$. Scale bars: 25 µm. Source data are available online for this figure.

multimer analysis (Jumper et al, 2021; Varadi et al, 2022). The model was based on a shortened CHC22 Hub trimer (residues 1278–1640) plus the full BAR domain (residues 202–404) of SNX5. The configurations produced by interaction modeling suggested the binding of the SNX5 BAR domain to the central trimerization domain (TxD, residues 1576–1640) within the Hub of CHC22 (Fig. 3D). The highest scoring model predicted that the SNX5 BAR domain binds on top of the TxD, interacting with all three heavy chain subunits within the trimer. This markedly differs from the way that clathrin light chains (CLCs) interact with the Hub region of CHC17 established by cryo-electron microscopy (Morris et al, 2019), X-ray crystallography (Wilbur et al, 2010) and mutagenesis (Chen et al, 2002). The CLCs align starting at approximately residue 1267 of CHC17 (Chen et al, 2002), near the bend (knee) of the triskelion leg, and extend along the proximal leg region to the TxD to interact at the sides of the TxD helix tripod (Morris et al, 2019), which projects to the opposite side to that predicted for SNX BAR binding. Repeating the AlphaFold simulation of the predicted interaction between CHC22 and SNX5 using the BAR domain of the SNX5 paralogue protein SNX6 (residues 205–406), generated models that were nearly identical to that of SNX5 binding to CHC22 (Fig. EV3A,B). As such, and in light of reported functional redundancy between SNX5 and SNX6, further molecular experiments focused on SNX5 alone.

The AlphaFold prediction was then tested in vitro using recombinantly expressed protein fragments (Figs. 3E,F and EV3C,D). First, the specificity of SNX5 for CHC22 clathrin compared to CHC17 was assessed. Full-length recombinant SNX5 protein tagged with glutathione-S-transferase (GST) was captured by the His-tagged Hub domain of CHC22 and not by the equivalent His-tagged CHC17 Hub bound to nickel affinity (Ni-NTA) beads (Fig. 3E). To refine the CHC22 binding site for SNX5 and validate the interaction with the TxD predicted by AlphaFold, His-tagged CHC22 Hub or His-tagged CHC22 TxD (residues 1520–1640) were bound to Ni-NTA beads and both were shown to capture GST-SNX5 (Fig. 3F), demonstrating that the CHC22 TxD is sufficient to bind SNX5 and supporting the AlphaFold prediction.

The AlphaFold modelling (Fig. 3D) further suggested that the SNX5 BAR domain interacts with CHC22 TxD using the same binding interface that has been predicted by structural homology modeling (Simonetti et al, 2019) for SNX5 binding to SNX1/2 in the ESCPE-1 complex (Fig. 4A). Overlaying the two models predicted steric hindrance between ESCPE-1 and CHC22-SNX5 interactions (Fig. 4B), suggesting that these two SNX5 complexes are mutually exclusive. To investigate competition between the two interactions, recombinant SNX1 was purified and incubated in increasing amounts with GST-SNX5 immobilized on glutathione beads (Fig. EV3C,D). The binding of purified CHC22 TxD binding to SNX1-associated SNX5 was then assessed (Fig. 4C). Concentration-dependent reduction of CHC22 TxD binding to SNX5 was observed when SNX1 was present (Fig. 4D), in agreement with SNX5 binding to CHC22 TxD using the same interface that binds SNX1 in the ESCPE-1 complex.

Further addressing the prediction that SNX5-CHC22 and SNX5-SNX1 interactions depend on the same interface, a previously characterized SNX5 mutant was assessed for binding to the CHC22 TxD. The mutant tested was a phosphomimetic point mutant (S226E) in the SNX5 BAR domain, which was reported to abolish SNX1/2 interactions (Itai et al, 2018; Simonetti et al, 2019), and loss of binding to SNX1 was confirmed (Fig. EV3E). When increasing amounts of the wild-type (WT) SNX5 BAR domain and the mutant SNX5 BAR domain were immobilized on beads, they both bound the CHC22 TxD, but binding by the S226E mutant was reduced compared to binding by the WT SNX5 BAR domain across a range of concentrations (Fig. 4E,F). Collectively, these data indicate that SNX1 and CHC22 interact with the same interface of SNX5, and compete with each other for this interaction.

## SNX5 links CHC22 to the ERGIC by binding p115

Previous work identified the ERGIC tether p115 in a complex with CHC22 and not with CHC17 (Camus et al, 2020). The reduced PeriMI of CHC22 and reduced p115 interaction detected by PLA in ΔSNX5/6 cells (Fig. 1C,E), as well as the correlation of CHC22

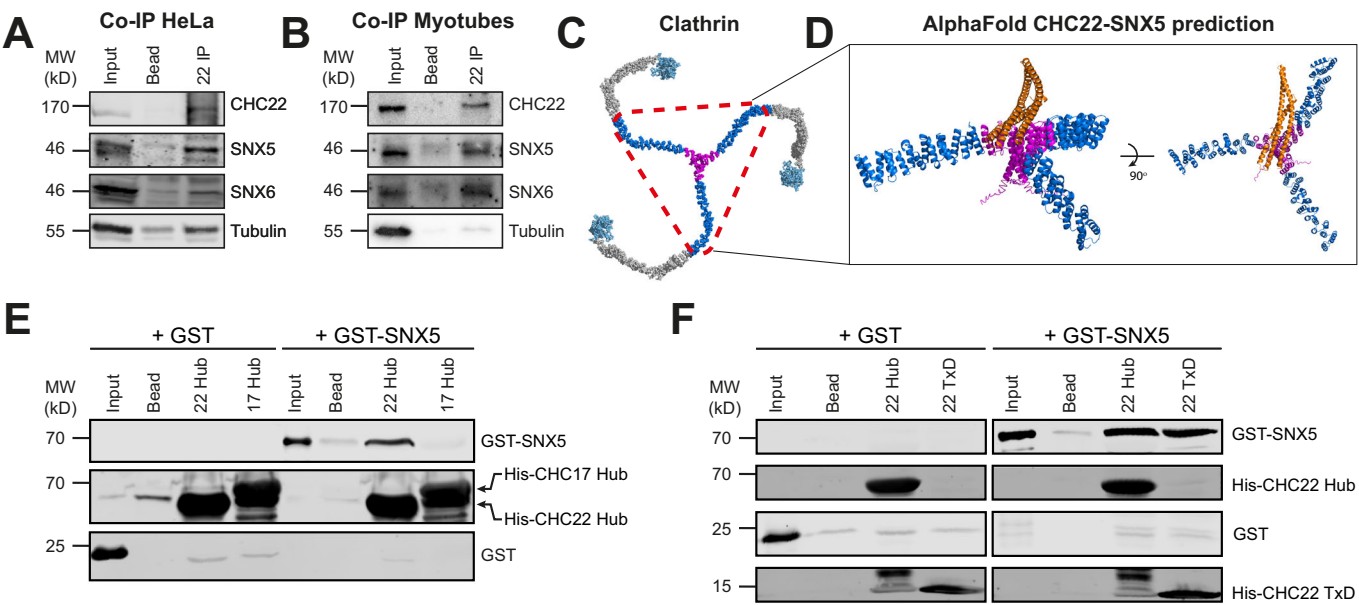

**Figure 3. SNX5 binds directly to CHC22 at the trimerization domain.**

(A, B) Representative immunoblot ($n = 4$–$6$) of immunoprecipitates (IP) of CHC22, from control HeLa cells (A) and differentiated human skeletal muscle cell line AB1190 (Myotubes) (B). Samples were immunoblotted for CHC22, SNX5, SNX6, and tubulin. Lanes show Input (5%), bead-only (no antibody) control (Bead) and CHC22 immunoprecipitate (22 IP). The migration positions of MW markers are indicated at the left in kilodaltons (kD). (C) The structure of a single clathrin triskelion formed from three clathrin heavy chains with the Hub region indicated by red dotted triangle and the trimerization domain (TxD), magenta, proximal leg region (dark blue), knee and distal leg regions (grey) and terminal domain (light blue), based on PDB 3IYV (Fotin et al, 2004). (D) AlphaFold-generated model of the interaction between CHC22 Hub (blue proximal leg residues 1278–1519 and magenta TxD residues 1520–1640) and the BAR domain of SNX5 (residues 202–404, orange). Two angles of view are displayed. (E) Representative immunoblot ($n = 3$–$5$) of in vitro binding of full-length GST-SNX5 (right lanes) or GST alone (left lanes) to CHC22 Hub (22 Hub) or CHC17 Hub (17 Hub) immobilized on Ni-NTA beads. Purified protein input (2%) and bead-only (no Hub immobilized) control (Bead) are shown for each prey. Samples were immunoblotted for SNX5, His-tag or GST with detected proteins indicated at the right. The position of MW markers is indicated at the left in kD. (F) Representative immunoblot of in vitro binding of full-length GST-SNX5 (right lanes) or GST alone (left lanes) to CHC22 Hub or CHC22 TxD immobilized on Ni-NTA beads ($n = 3$). Purified protein input (2%) and beads without CHC22 fragment added (Bead) are shown for each prey. Samples were immunoblotted for SNX5, His-tag or GST with detected proteins indicated at the right. The position of MW markers is indicated at the left in kD. Source data are available online for this figure.

PeriMI with SNX5/6 levels (Figs. 1 and 2) led to the question of whether p115 is involved in SNX5/6-mediated CHC22 recruitment. Immunoprecipitation of CHC22 from control parental HeLa cells compared to ΔSNX5/6, ΔSNX1/2, or ΔVPS35 cells showed that co-precipitation of p115 was lost in the ΔSNX5/6 cells, but present in the others (Fig. 5A,B). This suggested that SNX5/6 contributes to the interaction between CHC22 and p115.

Loss of CHC22 localization at the ERGIC has been observed following siRNA-mediated knockdown (KD) of p115 (Camus et al, 2020). To assess the role of SNX5/6 in CHC22 recruitment by p115, expression of SNX5 was tested for rescue of this phenotype following siRNA-mediated p115 depletion (Fig. 5C–E), with depletion efficacy verified (Fig. EV4A). Consistent with previous observations (Camus et al, 2020), the CHC22 PeriMI was reduced in control HeLa cells following p115 KD compared to cells treated with control non-targeting siRNA (control). The distribution of GM130 was also altered by p115 KD (Fig. 5C–E). In ΔSNX5/6 cells (Δ), CHC22 PeriMI was further (slightly) reduced by p115 KD (Fig. 5C, third row), D). However, in the p115-depleted ΔSNX5/6 cells (Δ), CHC22 perinuclear localization could not be restored by transfection of FLAG-tagged SNX5 (Fig. 5C (bottom row), D) as observed when p115 was present (Fig. 1J,K). This contrasts with the perinuclear localization of GM130, which was partially restored by transfection of SNX5 into p115-depleted ΔSNX5/6 cells (Fig. 5C

(bottom row), E). Together, these findings demonstrate that SNX5's role in CHC22 localization is dependent on p115 and highlights the differences in the molecular interactions of p115 with CHC22 and GM130.

In order to further explore the link between p115 and SNX5, in vitro pulldowns were utilized to investigate potential protein-protein interaction. Purified recombinant full-length GST-SNX5 was found to bind directly in vitro to recombinant full-length His-p115 immobilized on Ni-NTA beads, with no binding to a bead-only control or by GST alone (Fig. 5F). Thus, SNX5 acts as a direct molecular bridge between CHC22 and p115, contributing to CHC22 localization.

## An isotype-specific patch in the CHC22 N-terminal domain additionally mediates p115 binding

Canonical CHC17 clathrin is recruited to membranes by interactions with adaptor proteins at the N-terminal domain (TD) (Chen et al, 2020; Lemmon and Traub, 2012). The membrane recruitment role identified for SNX5/6 raises the question as to whether the SNX5-p115 interaction is both necessary and sufficient for CHC22 recruitment, or whether interactions mediated by the TD are also involved in CHC22 localization. To investigate this, a CHC22 mutant lacking the TD (residues 1–330) (ΔTD) was created and tagged with super-folded

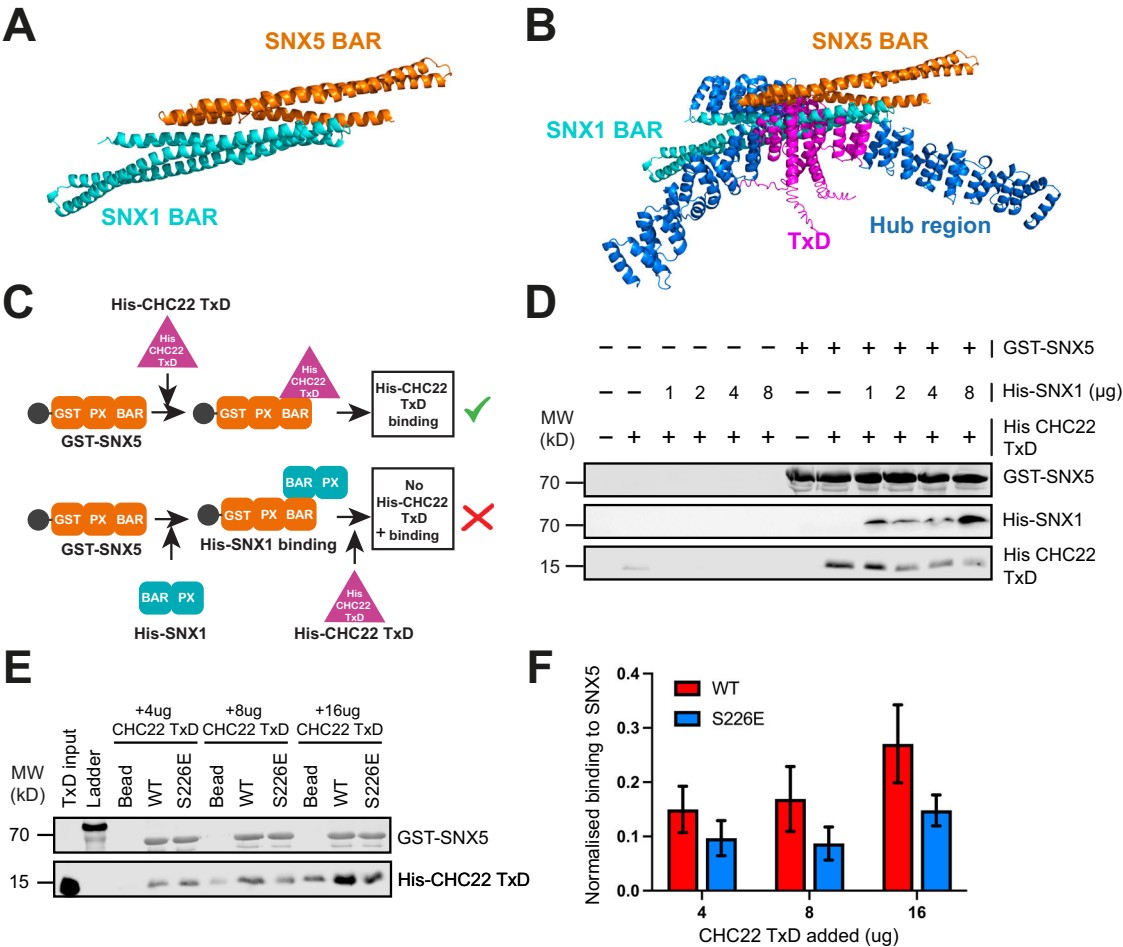

**Figure 4. CHC22 TxD and SNX1 compete for binding to SNX5 at the interface where SNX5 participates in the ESCPE-1 complex.**

(A) AlphaFold-generated model of the SNX1-SNX5 BAR domains interaction in the ESCPE-1 complex (SNX1 BAR residues 299–516 in cyan, SNX5 BAR residues 203–399 in orange). (B) Overlay of SNX5-SNX1 BAR domain heterodimer with SNX5 BAR bound to the CHC22 Hub as predicted by AlphaFold (CHC22 proximal legs in blue, TxD in magenta). (C) Schematic cartoon of assay testing whether SNX1 binding to SNX5 can compete with TxD binding to SNX5. Full-length GST-tagged SNX5 was bound to beads and exposed first to full-length His-tagged SNX1, then His-tagged CHC22 TxD was added and its binding was detected by immunoblotting with antibody against CHC22 TxD. (D) Representative immunoblot (n = 3) of the in vitro competition assay shown in (C), testing effects of increasing amounts (μg) of His-SNX1 added to GST-SNX5 on His-CHC22 TxD binding. The left six lanes show background binding when no GST-SNX5 was added to the beads. Samples were immunoblotted for GST, His-tag, or CHC22 with detected proteins indicated at the right. The position of MW markers is indicated at the left in kilodaltons (kD). (E) Representative immunoblot of the in vitro binding of purified His-CHC22 TxD (22 TxD) to immobilized full-length GST-SNX5 wild-type and phosphomimetic S226E mutant. Increasing amounts of His-CHC22 TxD, ranging from 4 to 16 μg, were added to bead-only control (Bead), GST-SNX5 WT (WT), and GST-SNX5 S226E (S226E). Samples were immunoblotted for SNX5 or CHC22 and detected proteins indicated at the right. His-CHC22 TxD input (0.1 μg) is shown on the far left and the position of MW markers is indicated in kD. (F) Quantification of the binding signals of His-CHC22 TxD to GST-SNX5 WT (red) and S226E (blue), as shown in (E). The TxD binding signals were calculated by subtracting the value of the appropriate bead-only condition and subsequently normalized to the His-CHC22 TxD input signal. Error bars display the standard error of the mean (n = 8). Source data are available online for this figure.

GFP (sfGFP). This protein was expressed in HeLa cells by transfection and its localization compared to a wild-type (WT) sfGFP-tagged CHC22 control (Fig. 6A,B). The WT sfGFP-CHC22 localized to the perinuclear region overlapping with p115 (Fig. 6A, middle). By contrast, the ΔTD CHC22 construct had no discrete localization pattern but was present throughout the cytoplasm (Fig. 6A, bottom). Quantification of CHC22-p115 colocalization (Fig. 6C) showed sfGFP-ΔTD-CHC22 had a significantly lower p115-colocalization coefficient (P = 0.015, Fig. 6C) compared to WT sfGFP-CHC22. This indicated that the CHC22 TD is also required for correct CHC22 localization in addition to the SNX5-p115 interaction, and that the latter is not sufficient.

To assess whether the CHC22 TD interacts with p115, GFP nano-trap reagents were used to isolate GFP-tagged WT-CHC22, ΔTD-CHC22 and WT-CHC17 from lysates of HeLa cells transiently transfected with the GFP constructs. Immunoblotting the isolates for p115 (Fig. 6D) showed that, relative to WT CHC22, the ΔTD CHC22 displayed reduced binding to p115, though not as low as CHC17 (Fig. 6E). To establish whether p115 directly binds to the CHC22 TD, TD fragments of CHC22 and CHC17 (residues 1–364 with His-SUMO tags) were purified (Fig. 6F) and tested for binding (Fig. 6G, H) to full-length p115 (GST-p115) and to the hinge-ear fragment of GGA2, a clathrin adaptor known to bind both CHC22 and CHC17 (GST-GGA2-HE) (Vassilopoulos et al,

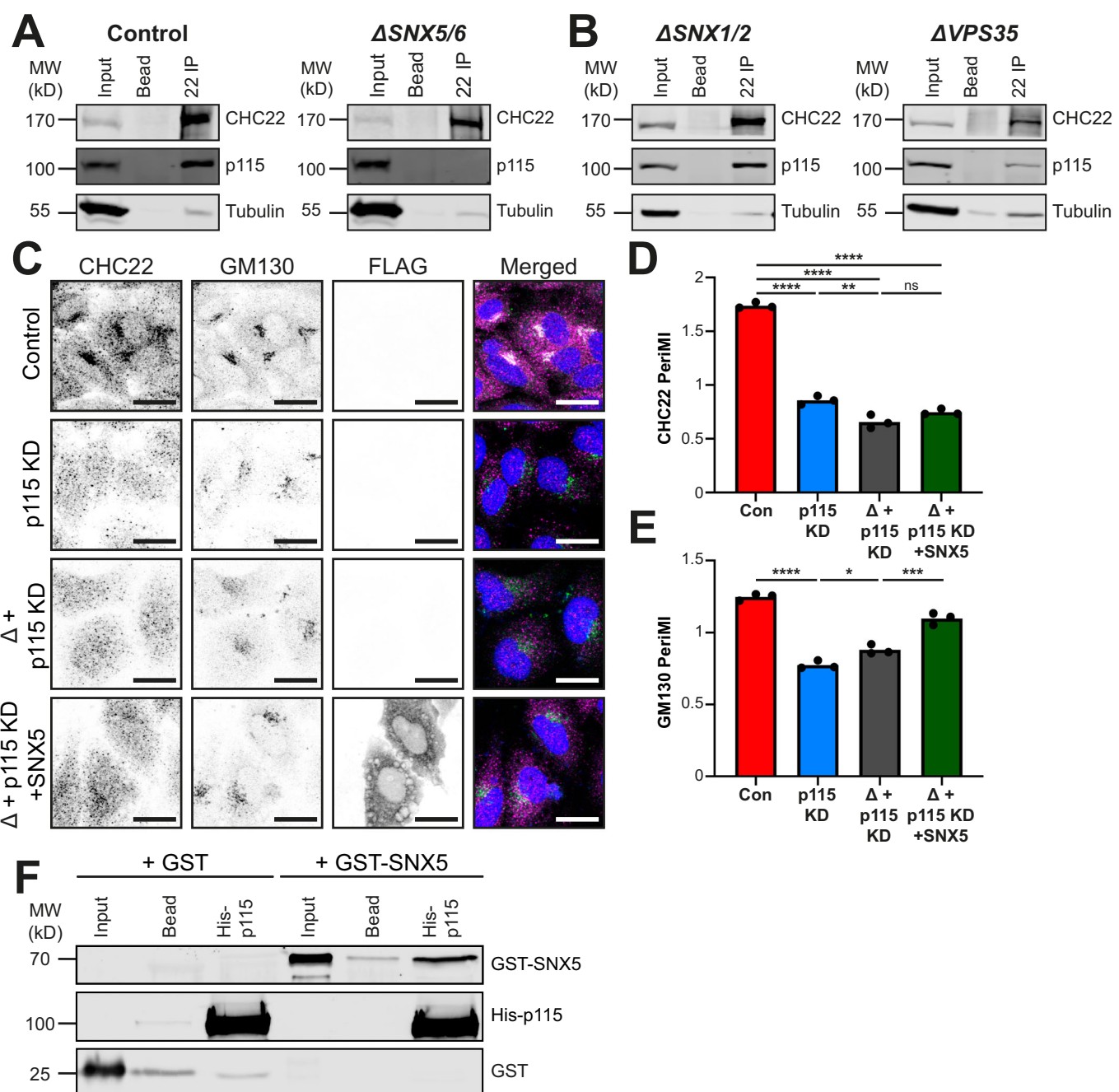

2009; Zhu et al, 2001), immobilized on glutathione beads. Immunoblotting shows that CHC22 TD directly binds both the GGA2 fragment and p115 (Fig. 6G), while the CHC17 TD does not interact with p115, only with the GGA2 fragment (Fig. 6H). This binding data is supported by previous co-immunoprecipitation studies, which showed that, in cells and muscle, GGA2 interacts with both CHC17 and CHC22 clathrin, but p115 only complexes with CHC22 (Camus et al, 2020; Dannhauser et al, 2017; Vassilopoulos et al, 2009).

To better understand CHC22 binding specificity for p115 in the context of known adaptor binding sites on CHC17, the sequence conservation between the TDs of CHC17 and CHC22 was analyzed.

All of the previously mapped CHC17 residues involved in adaptor binding sites were conserved between the two TDs (Fig. 7A). This suggested that either subtle structural perturbations in these binding sites alter their specificity, and/or that additional binding sites unique to CHC22 are responsible for mediating its specific localization. To search for potential unique binding sites, regions of the CHC22 TD that are divergent in sequence between CHC17 and CHC22 were identified on a homology model, derived from a CHC17 TD crystal structure (ter Haar et al, 1998), and colored based on CHC22 sequence conservation relative to CHC17 (Fig. 7B). This revealed a spatial clustering of nine divergent residues, which was not apparent from the primary amino acid sequence alone. To investigate the role of this

**Figure 5. SNX5 directly binds p115 and their interaction mediates CHC22 localisation.**

(A) Representative immunoblot ($n = 4$) of CHC22 immunoprecipitates from control parental HeLa (left) and $\Delta SNX5/6$ HeLa-derived lines (right) immunoblotted for CHC22, p115, and tubulin. Lysate input (5%), bead-only (no antibody) control (Bead) and CHC22 immunoprecipitate (22 IP) are shown for each cell line. The migration positions of MW markers are indicated at the left in kilodaltons (kD). (B) Representative immunoblot ($n = 4$) of CHC22 immunoprecipitates from $\Delta SNX1/2$ HeLa-derived line (left) and $\Delta VPS35$ HeLa-derived line (right) immunoblotted for CHC22, p115, and tubulin. Lysate input (5%), bead-only (no antibody) control (Bead) and CHC22 immunoprecipitate (22 IP) are shown for each cell line. The migration positions of MW markers are indicated at the left in kD. (C) Representative images of control parental HeLa line transfected with non-targeting siRNA (control, top), control parental HeLa line transfected with siRNA targeting p115 (p115 KD, second row), $\Delta SNX5/6$ cells transfected with siRNA targeting p115 ($\Delta$ + p115 KD, third row), and $\Delta SNX5/6$ cells transfected with siRNA targeting p115 and FLAG-SNX5-encoding plasmid ($\Delta$ + p115 KD + SNX5, bottom). Cells were immunolabeled for CHC22 (magenta in merged), GM130 (green in merged) and FLAG (not shown in merged for clarity) with overlap shown in white and DAPI-stained nuclei (blue) in merged. Scale bars: 25 µm. (D, E) Quantification of PeriMI for CHC22 (D) ($P < 0.0001$ (****), $P = 0.0016$ (**), $P = 0.1091$ (ns)) and GM130 (E) ($P < 0.0001$ (****), $P = 0.0163$ (*), $P = 0.0002$ (***)) for HeLa cells transfected with non-targeting siRNA (Con), HeLa cells transfected with p115 siRNA (p115 KD), $\Delta SNX5/6$ cells transfected with p115 siRNA ($\Delta$ + p115 KD), and $\Delta SNX5/6$ cells transfected with p115 siRNA and FLAG-SNX5-encoding plasmid ($\Delta$ + p115 KD + SNX5) as shown in (C). Each bar represents the mean normalized PeriMI value from three independent experiments, each individually depicted as dots (11–24 cells per condition, per experiment). Statistical analysis was performed using a one-way ANOVA with a Tukey post-hoc test. (F) Representative immunoblot ($n = 3$ independent experiments) of in vitro binding of full-length SNX5 (GST-SNX5) or GST to p115 (His-p115) on Ni-NTA beads. Samples were immunoblotted for SNX5, p115 or GST and detected proteins indicated at the right. Input (2%) and bead-only (no p115 containing lysate added) control (Bead) are shown for each prey. The position of MW markers is indicated at the left in kD. ns = not significant; *$P < 0.05$; **$P < 0.01$; ***$P < 0.001$; ****$P < 0.0001$. Source data are available online for this figure.

divergent patch in adaptor binding, each of the divergent residues in CHC22 TD was mutated to the equivalent CHC17 TD residue, creating a chimeric construct CHC22-TD-Chim (Chim) (Fig. 7B,C). In vitro binding to p115 and GGA2-HE immobilized on glutathione beads was then tested for both WT TD and CHC22-TD-Chim (Fig. 7D). For the CHC22-TD-Chim mutant, loss of binding to p115, relative to WT TD binding and comparable to background binding of GST alone, was observed (Fig. 7D). Both the WT and CHC22-TD-Chim bound to a recombinant fragment of GGA2 (Fig. 7D) demonstrating that the gross architecture of the TD was not adversely affected by mutating the CHC22 residues to their CHC17 equivalents, and strongly implicating the divergent patch in CHC22 TD specificity for p115. Overall, these data indicate that the CHC22 TD can bind directly to p115 independently of the SNX5/6-TxD interaction at the C-terminus. Furthermore, this interaction requires a divergent patch within the CHC22 TD, which is not shared by CHC17.

## Both determinants of CHC22 binding to p115 are required for formation of an insulin-responsive GLUT4 storage compartment

Having identified two molecular interactions by which CHC22 interacts with the ERGIC protein p115, the functional relevance of both interactions in CHC22-dependent membrane traffic was addressed. The most well-characterized CHC22 cargo is the glucose transporter GLUT4 (Camus et al, 2020; Fumagalli et al, 2019; Gould et al, 2020; Vassilopoulos et al, 2009), which CHC22 targets to an intracellular storage compartment (GSC) from which GLUT4 is released in vesicles in response to insulin signaling. The pathway is recapitulated in a HeLa cell line stably transfected to express a HA-GLUT4-GFP fusion construct (HeLa-G4). The HA tag is located on an exofacial loop to enable detection of GLUT4 on the cell surface, while the C-terminal GFP tag enables measurement of the total GLUT4 expressed (Martin et al, 2006). Due to HeLa cells expressing CHC22 at levels comparable to muscle tissue (Esk et al, 2010), and possessing all the other components of the CHC22 trafficking pathway, the HeLa-G4 cell line packages GLUT4 in a GSC and releases GLUT4 to the cell surface in response to insulin (Camus et al, 2020; Haga et al, 2011; Trefely et al, 2015).

The role of SNX5/6 in GLUT4 traffic was initially assessed by fluorescent recovery after photobleaching (FRAP) of the perinuclear storage compartment (Cheng et al, 2010; Eyster et al, 2006; Fumagalli et al, 2019; Greig and Bulgakova, 2021; Ishikawa-Ankerhold et al, 2012; Khalique et al, 2016; Kitamura and Kinjo, 2018) (Fig. 8A–C). In HeLa-G4 cells treated with control siRNA, the mobile fraction of HA-GLUT4-GFP in the GSC was approximately 50% (Fig. 8A, B). The recovery curve was best fit by a two-phase exponential model with the two components potentially representing diffusion and steady-state membrane traffic (Pincet et al, 2016; Sprague and McNally, 2005). In HeLa-G4 cells depleted of both SNX5 and SNX6 by siRNA-mediated KD (verified in Fig. EV4B,C), GLUT4-GFP recovery dynamics were slowed, with the mobile fraction reduced to 40%. Nonlinear regression analysis showed a significant difference in the recovery curves, using a sum of squares F-test to determine if the recovery curve differed between the datasets ($P < 0.0001$, Fig. 8A,B). By contrast, following siRNA treatment to deplete the Retromer component VPS35 (verified in Fig. EV4D), no significant change in perinuclear GLUT4-GFP dynamics was detected ($P = 0.061$, Fig. 8C). Thus, SNX5/6 are implicated in the general mobility dynamics and steady-state trafficking of GLUT4.

To examine the role of SNX5/6 in enabling the formation of a functional insulin-responsive GSC, the insulin-stimulated surface translocation of HA-GLUT4-GFP (Fig. 8D) was assessed after SNX5/6 KD by siRNA treatment (Fig. 8E). Translocation was determined on a per cell basis by flow cytometry (Camus et al, 2020; Fumagalli et al, 2019) and quantified as the median fluorescent intensity (MFI) of surface GLUT4 (HA signal) relative to the total GLUT4 (GFP signal) (Fig. 8E–G). Significant insulin-stimulated translocation of GLUT4 was lost in cells following SNX5/6 KD or CHC22 KD ($P = 0.358$, $P = 0.683$, respectively, Fig. 8E), while in control cells, an increase of 50% was observed between the basal and insulin-stimulated surface:total GLUT4 ratio (Fig. 8E), consistent with previous reports, and typical for GLUT4 translocation in muscle cells (Camus et al, 2020; Fumagalli et al, 2019). Furthermore, previous work has demonstrated that this effect is specific to CHC22 as depletion of CHC17 does not similarly perturb the ability of GLUT4 to translocate after insulin stimulation (Camus et al, 2020; Fumagalli et al, 2019). By contrast, when either SNX5 or SNX6 were depleted alone, GLUT4-GFP translocation was still observed ($P = 0.002$ and $P = 0.012$, respectively, Fig. 8E), supporting functional redundancy of SNX5 and SNX6 in this pathway. For comparison, VPS35 KD did not impair

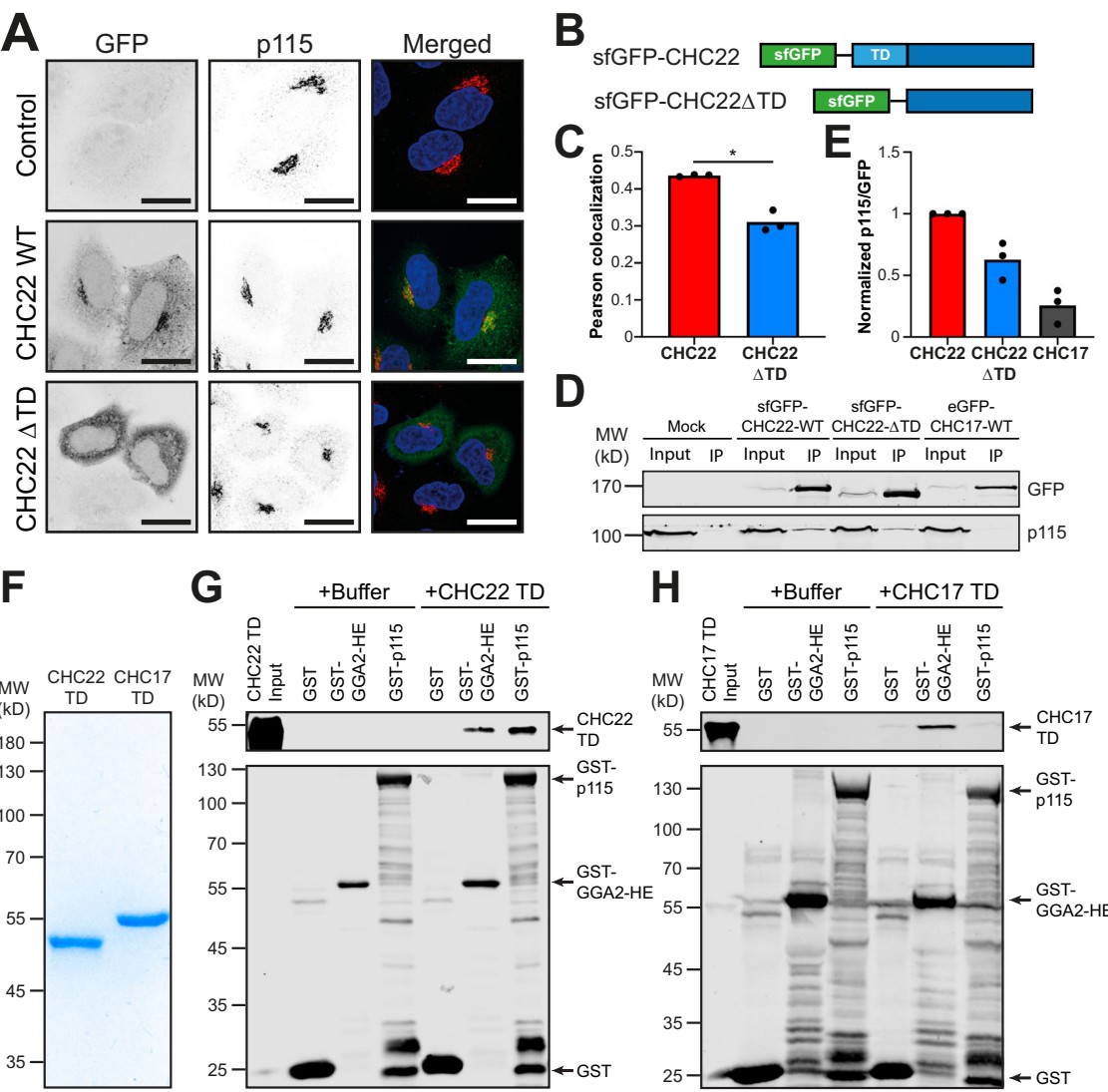

**Figure 6. The N-terminal domain (TD) of CHC22 is also required for correct localisation and binds p115 directly.**

(A) Representative images of control parental HeLa cells mock-transfected (Control-top), HeLa cells transfected with sfGFP-CHC22 wild-type expression plasmid (WT-middle), and HeLa cells transfected with sfGFP-ΔTD-CHC22 expression plasmid (ΔTD-bottom). Cells were imaged for GFP (green in merged) and immunolabeled for p115 (red in merged) with red-green overlap in yellow and DAPI-stained nuclei (blue) in merged. Scale bars: 25 μm. (B) A schematic of the encoded CHC22 constructs showing the location of the N-terminal sfGFP tag and the deletion of the TD (bottom). (C) Quantification of Pearson's colocalization for CHC22 and p115, in cells expressing full-length sfGFP-CHC22 (CHC22) or sfGFP-ΔTD-CHC22 (CHC22 ΔTD). Each bar represents the mean Pearson's colocalization value from three independent experiments, each individually depicted as dots (14–25 cells per condition, per experiment). Statistical analysis was performed using a two-tailed Student's *t* test with Welch's correction (*P* = 0.0154). (D) Representative immunoblot (*n* = 3) of anti-GFP immunoprecipitates (GFP nanotrap beads) from HeLa cells mock-transfected (Mock) or transfected with CHC22 WT (sfGFP-CHC22-WT), CHC22ΔTD (sfGFP-CHC22-ΔTD) or CHC17 WT (eGFP-CHC17-WT). Lysate input (5%) and immunoprecipitate (IP) are shown for each condition. Samples were immunoblotted for GFP or p115 as indicated at the right. The migration positions of MW markers are indicated at the left in kD. (E) Intensity of the p115 signal for each IP in (D), normalized relative to the GFP signal (p115/GFP) (*n* = 3 independent experiments). (F) Coomassie-stained SDS-PAGE gel of the purified CHC TDs (CHC22 TD or CHC17 TD). 1 μg of protein loaded in each lane. (G) A representative immunoblot of in vitro binding of His-SUMO-tagged CHC22 terminal domain (CHC22 TD) to GST only (GST), the clathrin-binding fragment of GGA2 tagged with GST (GST-GGA2-HE) and full-length GST-p115 bound to glutathione beads. CHC22 TD input (0.4%) is far left and background samples with only Buffer added (no TD) are shown. Samples were immunoblotted for the His-tag (top) or GST (bottom). The migration position of MW markers is indicated at the left in kD (*n* = 3). (H) A representative immunoblot (*n* = 3) of in vitro binding of His-SUMO CHC17 terminal domain (CHC17 TD) to GST, GST-GGA2-HE and full-length GST-p115 immobilized on glutathione beads. CHC17 TD input (0.4%) is far left and background samples with only Buffer added (no TD) are shown. Samples were immunoblotted for the His-tag (top) or GST (bottom). The migration position of MW markers is indicated at the left in kD. Note, due to differing background binding of CHC17 and CHC22 TDs to GST alone, pulldowns in (G, H) were carried out in different buffers (see methods).
*P < 0.05. Source data are available online for this figure.

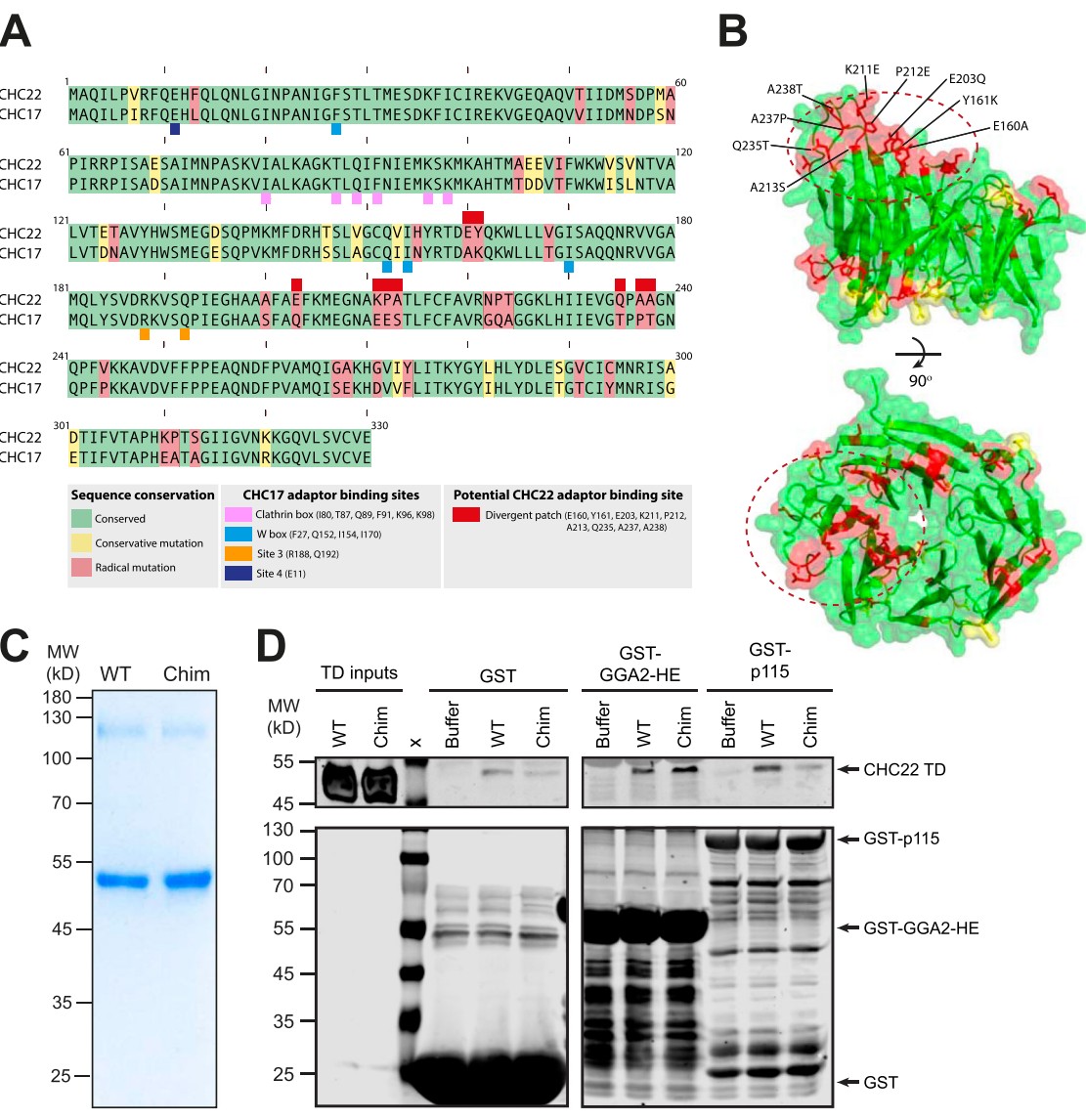

**Figure 7. Divergent N-terminal domain (TD) residues are responsible for direct CHC22 recruitment by p115.**

(A) Sequence homology analysis between human CHC22 and CHC17 TDs. Regions of shared sequence for residues 1–330 are shown in green, yellow represents conservative amino acid substitutions, while red denotes radical amino acid differences between CHC22 and CHC17. Colored blocks above and below the protein sequences denote residues involved in the known adaptor-binding sites of CHC17 and in the predicted divergent patch on the CHC22 TD, as indicated in the key. (B) A model of the CHC22 TD based on the solved CHC17 TD crystal structure (PDB: 1BPO) showing the positioning of the patch of divergent residues identified from comparison with the CHC17 TD in (A) (all differences with CHC17 color-coded in red and yellow as above with surface cluster of nine non-conservative differences denoted by dashed circle). (C) Coomassie-stained SDS-PAGE gel of the purified wild-type CHC22 TD (WT) and Chimeric CHC22 TD (Chim) with the nine patch residues mutated to those in CHC17. 1 μg of protein loaded in each lane with molecular weight (MW) marker migration positions indicated in kilodaltons (kD) at the left. (D) A representative immunoblot (n = 3) of in vitro binding of purified His-SUMO-tagged CHC22 TD fragments (WT or Chim) to GST-only, the purified clathrin-binding fragment of the adaptor GGA2 (GST-GGA2-HE) and full-length p115 (GST-p115) immobilized on glutathione beads. Inputs of the CHC22 TD constructs (0.4%) and the MW ladder marker lane (x) are shown along with background buffer-only signal (Buffer, no TD added) for each immobilized bait. Samples were immunoblotted for the His-tag (top) or GST (bottom). The position of MW markers is indicated at the left in kD. The section on the left (comprising the TD inputs and the GST) is transferred from the same gel as the section on the right (GST-GGA2-H-E and GST-p115) and the membrane has been cropped in the middle for clarity. Source data are available online for this figure.

HA-GLUT4-GFP translocation in response to insulin (*P* = 0.05, Fig. 8E).

To assess the functional role of the CHC22 TD-p115 interaction in GLUT4 translocation, HeLa cells expressing HA-GLUT4-GFP were depleted of CHC22 by siRNA-mediated KD and tested for rescue of GLUT4-GFP translocation by transfection with siRNA-resistant WT CHC22 or ΔTD CHC22 (Fig. 8F,G). In these cells, transfection with

WT CHC22 rescued the insulin-stimulated translocation of GLUT4-GFP (*P* = 0.049, Fig. 8G), while transfection with the siRNA-resistant ΔTD CHC22 construct showed no significant change in surface GLUT4-GFP after insulin stimulation (*P* = 0.79, Fig. 8G). Together, these functional assays indicate that both the CHC22 TD and SNX5/6 interactions are required to enable CHC22 activity, which is a prerequisite for the formation of the perinuclear GLUT4 storage

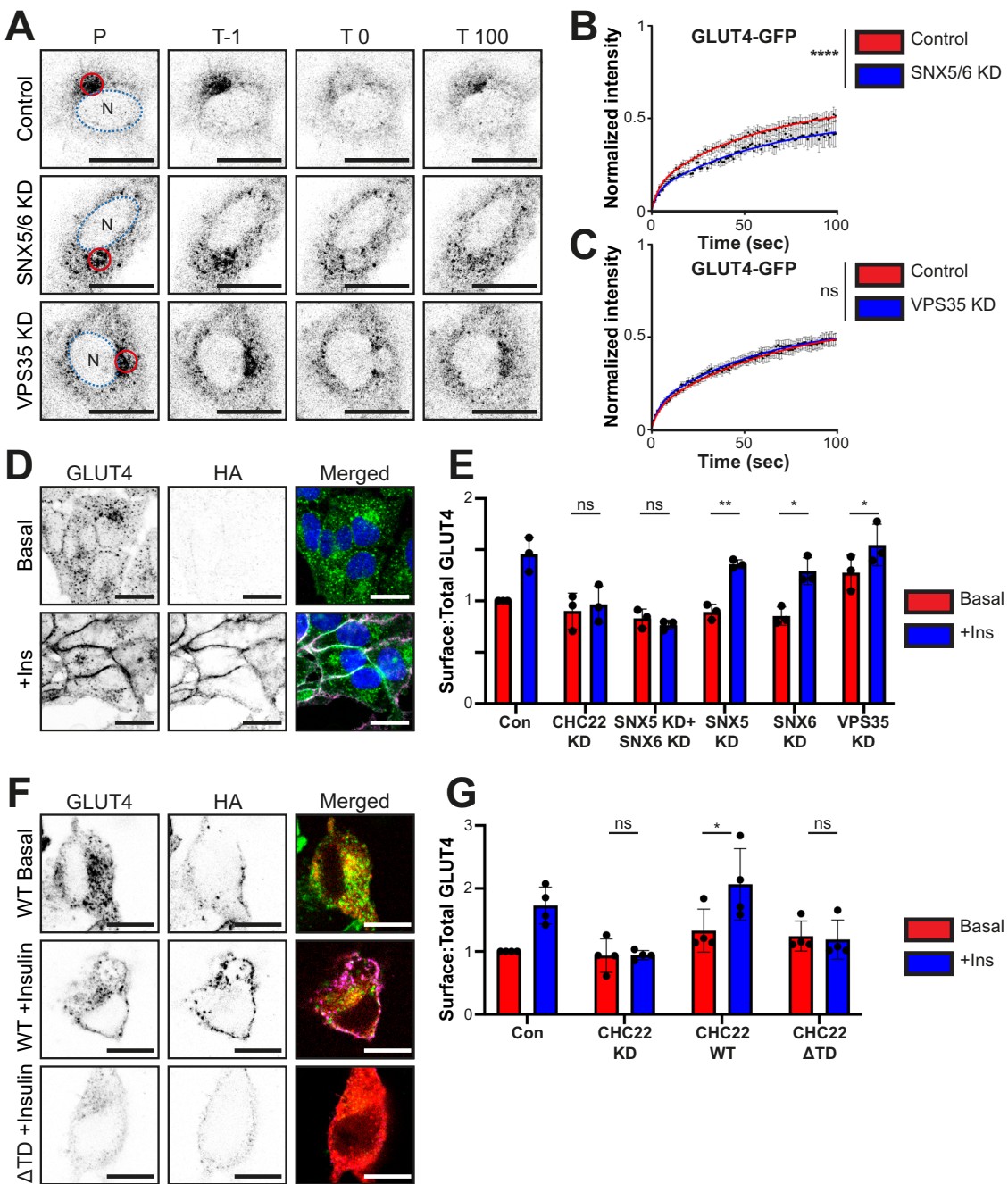

compartment, and that neither interaction can sustain CHC22 function without the other.

## Discussion

This study has characterized the molecular mechanism by which CHC22 clathrin is specifically recruited to the early secretory pathway, where it targets GLUT4 to the GSC. We have demonstrated that CHC22 recognizes the ERGIC tether p115 via direct binding at the N-terminal domain and indirectly at the C-terminus by interacting with SNX5. Disruption of either interaction impairs CHC22

recruitment and abrogates translocation of GLUT4 in response to insulin. This bipartite recruitment of CHC22 is fundamentally distinct from the recruitment mechanism of canonical CHC17 clathrin and provides an explanation of how CHC22 can function at a different intracellular location from CHC17. Our findings build on earlier results showing that CHC22 participates in two separate complexes with GLUT4; one comprising AP1 and GGA2 and one comprising p115 and IRAP. These complexes have been demonstrated to facilitate GLUT4 sorting to the GSC from the endosome and the ERGIC, respectively (Camus et al, 2020; Esk et al, 2010; Gould et al, 2020; Hosaka et al, 2005). The bipartite link between CHC22 and p115 described here highlights the role of p115 in the recruitment of CHC22

**Figure 8.  Both SNX5- and TD-mediated recruitment of CHC22 are necessary for function of the insulin-responsive GLUT4 storage compartment.**

(A–C) Dynamics of GLUT4–GFP measured by FRAP in the HeLa HA-GLUT4-GFP (HeLa G4) stably-transfected cell line. (A) A representative example of FRAP in HeLa G4 cells transfected with control non-targeting siRNA (Control), HeLa G4 cells transfected with siRNA targeting both SNX5 and SNX6 (SNX5/6 KD), and HeLa G4 cells transfected with siRNA targeting VPS35 (VPS35 KD). Panels in (A) show the position of the bleach spot (P, red circle) and the Nucleus (N, blue dashed ellipse) at the pre-bleach timepoint (T-1), bleach timepoint (T 0), and at the end timepoint (T 100). Time is in seconds. (B) Quantification of the HA-GLUT4-GFP fluorescence recovery post-bleaching between Control and SNX5/6 KD transfected HeLa GLUT4-GFP cells. Average recovery curves (mean ± SEM) and the best-fit curves (solid lines) are shown ($n = 3$, 7–11 cells per experiment, per condition). Analysis was performed using a nonlinear regression, with a sum of squares F-test ($P < 0.0001$). (C) Quantification of the HA-GLUT4-GFP fluorescence recovery post-bleaching between Control and VPS35 KD transfected HeLa GLUT4-GFP cells. Average recovery curves (mean ± SEM) and the best-fit curves (solid lines) are shown ($n = 3$, 8–11 cells per experiment, per condition). Analysis was performed using a nonlinear regression, with a sum of squares F-test ($P = 0.061$). (D) Representative images of control HeLa G4 cells without (Basal) or with insulin treatment (+ Insulin). Total GLUT4 imaged using GFP tag (green in merged), surface GLUT4 detected by live immunolabeling the exofacial HA tag (magenta in merged), with GFP overlap in white and DAPI-stained nuclei (blue) in merged. (E) Quantification of the HA-GLUT4-GFP surface-to-total ratio (HA:GFP) using median fluorescent signal (MFI). HeLa GLUT4-GFP cells were transfected with control non-targeting siRNA (Con) or transfected with siRNA (KD) targeting CHC22, SNX5 and SNX6, SNX5 only, SNX6 only, or VPS35, and then treated with vehicle only (Basal) or insulin (+ Ins). Bar height represents the mean value for all experiments in each condition. Each dot represents the normalized MFI value, relative to control, for all cells in each individual experiment. Statistical analysis was performed using a two-tailed Student's *t* test with Welch's correction between cells of the same experimental condition which were either vehicle only or insulin-treated ($P = 0.683$, $P = 0.358$, $P = 0.002$, $P = 0.012$, $P = 0.05$ from left to right, $n = 3$, 10,000 cells per condition, per experiment). (F) Representative images of control HeLa G4 cells depleted for endogenous CHC22 by siRNA treatment then transfected to express siRNA-resistant wild-type CHC22-mApple (WT) or siRNA-resistant CHC22-mApple with a terminal domain deletion (ΔTD) without (Basal) or with insulin treatment (+ Insulin). Total GLUT4 was imaged using GFP tag (green in merged), CHC22-mApple constructs were imaged using Apple tag (red in merged), and surface GLUT4 was detected by live immunostaining the exofacial HA tag (magenta in merged). (G) Quantification of the HA-GLUT4-GFP surface-to-total ratio (HA:GFP) using median fluorescent signal (MFI). HeLa GLUT4-GFP cells were either transfected with control non-targeting siRNA (Con) or transfected with siRNA (KD) targeting CHC22 alone (CHC22 KD) or combinations of siRNA targeting CHC22 plus constructs encoding siRNA resistant CHC22 WT or CHC22 ΔTD. These were then treated with vehicle only (Basal) or insulin (+ Ins). Bar height represents the mean value for all experiments in each condition. Each dot represents the normalized MFI value, relative to control, for all cells in each individual experiment. Statistical analysis was performed using a two-tailed Student's *t* test with Welch's correction between cells of the same experimental condition which were either vehicle only or insulin-treated ($P = 0.961$, $P = 0.049$, $P = 0.79$ from left to right, $n = 4$, 20,000 cells per condition, per experiment). ns = not significant; *$P < 0.05$; **$P < 0.01$; ****$P < 0.0001$. Scale bars: 25 μm. Source data are available online for this figure.

to the ERGIC and the equal importance of this GLUT4 traffic pathway in establishing the human GSC relative to the endosomal recapture pathway.

We can now model a bipartite mechanism whereby CHC22 is recruited to the early secretory pathway (Fig. 9). The SNX5/6 interaction at the CHC22 C-terminal domain may act as first stage (Fig. 9, step 1) in a sequential interaction that then enables the CHC22 N-terminal domain to dock onto p115, in a more conventional adaptor-like interaction, typical of CHC17 (Fig. 9, step 2). This idea is supported by the fact that separate loss of each interaction results in reduction of CHC22 localization to the ERGIC, indicating non-redundancy of both contacts. As p115 contains a long tail domain (~45 nm) (Striegl et al, 2009), it is topologically possible that p115 can sustain two interactions with CHC22 spanning the thickness of the clathrin lattice, reaching the SNX5/6 present on the upper surface of the TxD and the TD oriented internal to the lattice. The flexible hinge region of the AP2 adaptor complex has been shown to extend over a similar range to link the clathrin lattice to the plasma membrane in CHC17-coated vesicles (Kovtun et al, 2020).

Demonstrating the participation of SNX5/6 in p115 binding further verifies SNX5 as a CHC22 partner (Towler et al, 2004), and also defines a function for SNX5/6, independent of participation in the endosomal recycling ESCPE-1 complex. This mode of interaction between clathrin and sorting nexins has not been previously documented. While sorting nexin 9 (SNX9) and sorting nexin 4 (SNX4) are known interactors of CHC17 at the plasma membrane, they primarily facilitate endocytosis by remodeling actin and enabling scission of vesicles (Skanland et al, 2009; Soulet et al, 2005), and CHC17 is not involved in the formation of SNX-dependent carriers (McGough and Cullen, 2013). Here we show that CHC22 binds SNX5 and requires only SNX5/6 and not SNX1/2 (its ESCPE-1 partners) to associate with ERGIC membranes. Apparently CHC22 does not use the SNX5/6 BAR domain for its potential capacity to induce membrane curvature, as

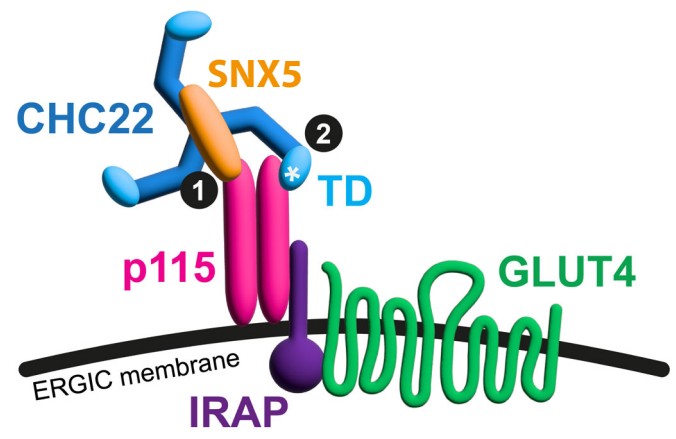

**Figure 9.  Model of the bipartite mechanism for CHC22 recruitment to ERGIC membranes.**

Diagram of how sequential interactions of CHC22 with p115 could recruit CHC22 to membranes of the Endoplasmic Reticulum-Golgi intermediate compartment membranes (ERGIC) for GLUT4 sequestration. The trimerization domain of CHC22 binds SNX5, which in turn binds p115 (interaction denoted by 1). This positions the N-terminal domain of CHC22 to directly bind p115 through a unique patch (asterisk) that diverges from the CHC17 sequence (interaction denoted by 2). Disruption of either interaction abrogates CHC22 recruitment and function. The ERGIC membrane is distinguished by the localization of p115. Through the interaction of p115 with IRAP, this complex (described previously) is linked to the capture of GLUT4 and its sorting to an insulin-responsive intracellular compartment upon assembly of the CHC22 coat at the ERGIC membrane.

this domain directly binds to the CHC22 TxD (Fig. 3). Consistent with AlphaFold modeling, binding of SNX5 to CHC22 is competed by SNX1 and depletion of SNX1/2 did not reduce CHC22 localization with p115, nor was SNX1 present in CHC22 immunoprecipitates

(Fig. EV2). The increased CHC22 perinuclear recruitment following depletion of SNX1/2, as observed for overexpression of SNX5, suggested that SNX5/6 was released by SNX1/2 depletion making more available to recruit CHC22 (Fig. 2) and confirms lack of SNX1/2 interaction with CHC22. Overall, these findings indicate that CHC22 clathrin is repurposing a SNX-BAR protein. Prior evidence for SNX-BAR protein involvement in the early secretory pathway is limited, although recent reports suggest that SNX19 tethers and positions ER exit sites (Saric et al, 2021) and identify roles for SNX4 in autophagosome formation and for SNX5 in xenophagy, both of the latter also originating from ER/ERGIC membranes (Dong et al, 2021; Nemec et al, 2017; Wu et al, 2022). Our finding that SNX5 binds the ERGIC tether protein p115 expands this emerging view of SNX-BAR activity beyond the endosomal system.

It has been reported that assembly and disassembly of the ESCPE-1 complex are influenced by phosphorylation of SNX5/6 at the interface with SNX1/2 (Itai et al, 2018). Mutation of SNX5 to a phosphomimetic residue that abrogates SNX1/2 binding partially reduced SNX5 binding to CHC22 (Figs. 4E and EV3E), supporting the AlphaFold model suggesting the CHC22-SNX5 interaction uses an overlapping interface as predicted for SNX5 to bind SNX1/2. However, the extent of phosphomimetic binding inhibition is stronger for SNX1 than for CHC22 suggesting that the CHC22-SNX5 interface may be dependent on additional residues. Whether CHC22 binding is affected by phosphorylation and how its competition with SNX1/2 in forming the ESCPE-1 complex is regulated remains to be determined.

The TxD of CHC22 alone is sufficient to bind SNX5, in a manner distinct from the configuration of clathrin light chain (CLC) interaction with the Hub (C-terminal third) region of CHC17. Our AlphaFold modeling predicts that the CHC22-SNX5 interaction could extend 40 Å along the proximal leg of the CHC22 triskelion. By comparison, the CLC interaction with CHC17 extends from the side of the TxD along the triskelion leg to its characteristic bended knee (125 Å from the TxD) (Wilbur et al, 2010). Endogenous CHC22 immunoprecipitated from cells does not bind CLCs, and anti-CLC antibodies do not co-localize with CHC22 in cells, although CHC22-CLC interactions can be detected by yeast-two-hybrid assays (Dannhauser et al, 2017; Liu et al, 2001; Towler et al, 2004). The original mapping of SNX5 binding to the CHC22 Hub region by yeast-two-hybrid suggested that SNX5 might be a CLC substitute for CHC22 (Towler et al, 2004). In so far as SNX5 interacts with the TxD, it may provide the trimer-stabilizing function imparted by CLC to CHC17 (Ybe et al, 2007). Indeed, recent phylogenetic studies identifying CLCs of highly divergent sequences in protist eukaryotes (Santos et al, 2022) indicate that CLC equivalents are predicted to interact with the TxD of CHC17 equivalents, and that this may be the essential molecular requirement for a CHC-associated subunit. As SNX5/6 interacts principally with the TxD of CHC22 rather than spanning the proximal leg of the heavy chain to the knee where CLC affects CHC17 assembly dynamics (Redlingshöfer and Brodsky, 2021), one possibility is that SNX5/6 binding influences the CHC22 uncoating dynamics. CHC22 has markedly different uncoating properties to CHC17 (Dannhauser et al, 2017) and the truncation of the CHC22 TxD relative to CHC17, losing the recognition site for the uncoating ATPase (Rapoport et al, 2008), is a notable distinction between the two clathrin isoforms.

This study provides the first direct evidence of the CHC22 TD mediating adaptor interactions in vivo. This is consistent with the clathrin recruitment paradigm derived from CHC17, in which the TD comprises four separate binding sites that interact with a collection of well-studied adaptor proteins to recruit clathrin to distinct cellular compartments (Briant et al, 2020; Lemmon and Traub, 2012; Robinson and Bonifacino, 2001; Smith et al, 2017; Wood and Smith, 2021). As both CHC22 and CHC17 share some in vivo interactors (e.g., GGA2 and AP1), but are distinct in others (e.g., AP2) (Dannhauser et al, 2017; Esk et al, 2010; Liu et al, 2001), it was predicted that some divergence within the TD could account for CHC22 recruitment specificity. When the CHC22 TD was modeled on the known structure of CHC17 TD, a divergent patch of nine residues on the surface of the CHC22 TD was identified, while additionally all the known adaptor binding sites in CHC17 are conserved in the CHC22 TD (Fig. 7A,B). Notably, when the residues comprising the divergent patch were all mutated to the equivalent CHC17 residues, p115 binding was abolished, while binding of a GGA2 fragment was maintained. This observation indicates that p115 interaction with the identified patch, plus p115 binding to CHC22 via SNX5/6, enables CHC22 function at the ERGIC. It may be that the divergent CHC22 TD patch also influences CHC22 TD conformation to increase its specificity for the CHC17 adaptors AP1 and GGA2 and lose AP2 binding. Alternatively, there may be additional CHC22 interactions that enhance its function in endosomal sorting of GLUT4 and reduce AP2 binding in vivo accounting for lack of CHC22 function in uptake from the plasma membrane (Dannhauser et al, 2017; Esk et al, 2010; Liu et al, 2001).

Overall, a bipartite mechanism for interaction of the specialized CHC22 clathrin isoform with the p115 tether in the early secretory pathway, a localization distinct from canonical CHC17, was identified and characterized. We found that this dual interaction is functionally required to form a mature GSC from which GLUT4 can translocate to the cell surface upon insulin stimulation. In insulin-resistant human muscle, CHC22 accumulates at sites of GLUT4 over-sequestration (Vassilopoulos et al, 2009), so interference with CHC22 membrane recruitment by targeting either site of the CHC22-p115 interaction defined here could potentially alleviate aberrant CHC22 accumulation during insulin resistance. Several cargoes, including CFTR, αps1-integrin, and several types of nutrient transporters are known to bypass the Golgi from the early secretory pathway, as observed for the CHC22-mediated GLUT4 pathway (Camus et al, 2020; Dimou et al, 2022; Grieve and Rabouille, 2011; Schotman et al, 2008; Tveit et al, 2009). Though CHC22 expression in non-muscle tissues is low, the CHC22 recruitment mechanism of interaction with p115 defined in this study could be a broader feature of early secretory trafficking.

## Methods

### Cell culture

All cell lines were cultured at 37 °C in a 5% $CO_2$ atmosphere with 100% humidity. HeLa cells were grown in DMEM high glucose (Gibco) supplemented with 10% FBS (Gibco), 50 U/ml penicillin and 50 μg/ml streptomycin (Gibco). The HeLa lines used in this

study were as follows: Parental (Control for the null cell lines), SNX1/2 Null (Δ*SNX1/2*), SNX5/6 Null (Δ*SNX5/6*), VPS35 Null (Δ*VPS35*) (Simonetti et al, 2019), and HA-GLUT4-GFP (HeLa-G4) (Camus et al, 2020). The human myoblast cell line AB1190 was grown in complete Skeletal Muscle Cell Growth Medium (Promocell) with provided supplemental cocktail and 10% FBS (Gibco) to reach a 15% final supplement concentration (volume/volume) (Camus et al, 2020). Differentiation of the AB1190 myoblasts was induced when culture was near full confluence (> 90%) by changing to Skeletal Muscle Cell Differentiation Medium (Promocell) and culturing for a further 12–14 days with provided insulin supplement only. All cell lines used tested negative for mycoplasma infection.

## siRNA

Targeting siRNA was synthesized (Thermo Fisher Scientific) using the following DNA sequences for each target: CHC22 (5′-TCG GGC AAA TGT GCC AAG CAA-3′ and 5′-AAC TGG GAG GAT CTA GTT AAA-3′,1:1 mixture of siRNAs were used) (Camus et al, 2020), p115 (5′-AAG ACC GGC AAT TGT AGT ACT-3′) (Camus et al, 2020). For SNX5 and SNX6 the following siRNA sequences were used: SNX5 (5′-UUAGUUUCAGCCCGAAGCAUC-3'), SNX6 (5′- UUAUGAGGUAGACGACUAAAU-3') (Wassmer et al, 2007), VPS35 (ID 132357, AM16708) (Predesigned—Thermo Fisher Scientific). Nontargeting control siRNA (control) was *Silencer^tm* Negative Control No.5 (AM4642) (Predesigned—Thermo Fisher Scientific).

## Plasmid constructs

A previously-generated FLAG-SNX5 construct was used for rescue and SNX5 overexpression experiments (Towler et al, 2004). For the terminal domain (TD) experiments, sfGFP-CHC22 and GFP-CHC17-encoding plasmids were used (Esk et al, 2010) either in full-length or as a template for the creation of deletion mutants and chimeras. In brief, the plasmid encoding the full-length protein was used as a template to design primers to amplify all but the region desired to be deleted. These were generated and transformed into DH5α competent E. coli cells using the Q5 mutagenesis kit in accordance with the manufacturer's instructions (NEB). For FACS rescue experiments with WT-CHC22 or ΔTD-CHC22, the sfGFP-WT-CHC22 and sfGFP-ΔTD-CHC22 plasmids were used as a template for exchanging the fluorescent tag from sfGFP to mApple (mApple encoding donor plasmid was a gift from Dr Samantha J. Warrington, Lab of Prof David Strutt, University of Sheffield) a HiFi (NEB) reaction was used to exchange the sfGFP for mApple using overlapping primers, in accordance with the manufacturer's instructions.

## Antibodies

The primary antibodies in this study were used at a dilution of 1:200-1:500 for IF, for immunoblotting the primary antibodies were used at a dilution of 1:2000–1:5000 or concentration noted, as for immunoprecipitation. The primary antibodies used in this study include: cell line supernatant from mouse monoclonal anti-p115 (clone 7D1) (Waters et al, 1992), rabbit polyclonal anti-CHC22 (22283-1-AP, Proteintech), goat polyclonal anti-SNX5 (ab5983, Abcam), rabbit polyclonal anti-SNX5 (ab180520, Abcam), rabbit

polyclonal anti-SNX6 (ab247087, Abcam), goat polyclonal anti-VPS35 (ab10099, Abcam), goat polyclonal anti-GM130 (sc-16268, Santa Cruz), rabbit polyclonal anti-TfR (ab84036, Abcam), mouse monoclonal anti-tubulin (ab7291, Abcam), mouse monoclonal anti-HA (16B12, BioLegend), mouse monoclonal anti-FLAG (M2 no.1804. Sigma), mouse monoclonal anti-His, (MAB050H, R&D Systems), rabbit polyclonal anti-GST, (#2622, Cell Signaling).

Secondary antibodies were used as follows: For IF, species-specific secondaries conjugated to, Alexa Fluor 488, Alexa Fluor 555, or Alexa Fluor 647 (Thermo Fisher Scientific) were used at a dilution of 1:1000. For fluorescent western immunoblotting, secondaries conjugated to IRDye were used, anti-mouse 680RD and anti-rabbit 800CW (LI-COR Biosciences) at a dilution of 1:10,000. For SNX5 and SNX6 fluorescent immunoblotting the primary antibodies were directly conjugated with DyLight 800 dye using conjugation kit, in accordance with manufacturer's instructions (DyLight^TM, Thermo Fisher Scientific). For chemiluminescent western blotting, antibodies coupled to horseradish peroxidase; anti-rabbit, anti-goat (HRP, Bio-Rad) were used at a dilution of 1:5000.

## Transfection

HeLa cells were seeded at a density of 15,000 cell/cm² in either 24-well plates or 12-well plates on #1.5 glass coverslips (Thermo Fisher Scientific). The following day, cells were observed to ensure satisfactory attachment and confluency (50–70%) before proceeding. For transfection of siRNA, oligonucleotides were complexed with jetPRIME (Polyplus) in accordance with manufacturer's instructions and added to the cells to a final concentration of 20 nM for all targets. Cells were then returned to normal growth conditions and harvested for analysis or fixed for microscopy imaging 72 h later. Depletion was confirmed by immunoblotting. For transfection of DNA plasmids, cells were seeded at the same density and the following day, were transfected with plasmid DNA which was complexed with jetPRIME in a 1:2 mixture (DNA/jetPRIME) in accordance with manufacturer's instructions. For SNX5 and CHC22 plasmids, 0.5 μg of DNA was used per well in a 12-well dish. Cells were then returned to normal growth conditions and harvested for analysis or fixed for microscopy imaging 24 h later. For FACS experiments requiring both siRNA and DNA plasmid transfection, a combination of the two protocols was used: cells were seeded, transfected the following day with siRNA and 48 h later transfected again with plasmid DNA. Cells were harvested for analysis or fixed for microscopy imaging the following day at the 72 h post-initiation endpoint.

## Immunofluorescent staining

Cells were seeded on #1.5 glass coverslips (Thermo Fisher Scientific) in dishes and experimental procedures (transfection, insulin stimulation etc.) were performed. At the endpoint of experiments, cells were washed once with PBS, then fixed in 4% PFA for 20 min at room temperature (21 °C). After this, the cells were washed three times with PBS. Cells were then permeabilized in PBST (PBS with 0.2% Triton X-100) for 10 min, followed by blocking in PBST with 2% horse serum (Gibco) for 1 h at room temperature. For staining, cells were incubated with primary antibody (diluted in PBST) overnight at 4 °C. The following day,

cells were washed three times in PBS and incubated with secondary antibodies (diluted in PBST) for 1–2 h at room temperature. Cells were then washed once in PBS before addition of DAPI staining solution to label nuclei (DAPI at 1:10,000 in PBS) for 20 min. After this time, cells were washed a further three times before coverslips were mounted on microscope slides (Superfrost, Thermo Fisher Scientific) using 2 μl of Vectashield per coverslip (Vector Laboratories) and sealed with nail varnish.

## Proximity ligation assays (PLA)

Cells were prepared in the same manner as for immunofluorescent staining up to and including the overnight incubation with primary antibodies (see previous section). The following day the cells were washed three times with Buffer A (Duolink, Sigma-Aldrich) at room temperature and then incubated with species-specific secondary PLA probes (diluted in Duolink diluent solution, Sigma-Aldrich) for 1 h at 37 °C in a humidified chamber. Cells were then washed in Buffer A three times and incubated with ligation solution (Duolink, Sigma-Aldrich) for 30 min at 37 °C in a humidified chamber. After this time, the cells were washed in Buffer A and incubated with amplification solution (Red, Duolink, Sigma-Aldrich) for 1 h 40 min at 37 °C in a humidified chamber. The cells were then washed in Buffer B (Duolink, Sigma-Aldrich) after which, nuclei were stained with DAPI and then mounted on microscope slides (see previous section).

## Image acquisition and analysis

All samples were imaged using a Leica TCS SP8 inverted laser-scanning confocal microscope equipped with two high-sensitivity hybrid detectors and one photomultiplier. A 63x (1.40 NA) HC Plan-Apo CS2 oil-immersion objective was used with five laser lines. Fluorophores were sequentially excited at 405 nm (DAPI), 488 nm (GFP, Alexa Fluor 488), 543 nm (Alexa Fluor 555), 561 nm (Alexa Fluor 568), and 633 nm (Alexa Fluor 647). For fixed images, 8-bit images were acquired ($1024 \times 1024$ XY pixel aspect) without further digital magnification. For each image, a Z-axis sectional stack was acquired using the nuclear DAPI signal as a positional marker. This consisted of 9 sections with 0.5 μm intersectional spacing, corresponding to a total Z-axis length of 4 μm. The images were saved in the Leica binary image format (.lif) and exported for further processing. For analysis, the images were processed using a series of custom macros in Fiji (NIH). In brief, the images were sorted by number and channel and, for each individual channel, an average intensity projection of the Z-section was extracted, created, and sorted. For analysis of the perinuclear mean intensity (PeriMI), the DAPI channel was used to create two masks for each cell in each image: one of the original dimensions of the nucleus, the other dilated by a factor corresponding to an isotropic increase in radius of 4.5 μm. The two masks were applied to the average intensity projection of the channels to be quantified using a macro incorporating the 'Analyze particles' function in Fiji. This measured the area and total pixel intensity for the original and dilated areas for each channel, which was saved as a .csv file. These data were then processed using a custom script in R. In brief, the mean of the 'total pixel intensities' of the original masks was subtracted from the mean of the 'total pixel intensities' from the dilated mask to provide a mean 'total pixel intensity' value for the

region between the two masks, corresponding to the perinuclear area (masks generated from 8 to 29 cells per condition per experiment). This value was in turn normalized to the mean area of the original masks subtracted from the mean area of the dilated masks (the perinuclear area), generating a PeriMI measurement corresponding to the mean intensity for the perinuclear area. These perinuclear mean intensity values were exported as .csv for subsequent statistical analysis. To account for differences in fluorescence intensity between repeat experiments, for each, PeriMI results were normalized by the mean of all the images of all the genotype/conditions. For calculating the proximity ligation assay (PLA) puncta, the DAPI stained nuclei were counted for each image alongside processing of the PLA signal channel. In brief, the PLA signal channels were processed to remove background signal using thresholding and converted to a binary image. The puncta present in these binary images were then counted and the area of each particle was measured using the 'Analyze particles' function in Fiji. To identify particles containing multiple puncta, which could not be sufficiently resolved by confocal microscopy or the application of 'Watershed' segmentation functions, the area of each particle identified in each image was grouped by size in a frequency histogram using the minimal puncta size for the bin intervals. To correct for multiple puncta in a single identified particle, the number of particles in each bin interval was multiplied by the corresponding area bin value. The sum of these corrected puncta number values was then divided by the corresponding nuclei number to generate the final puncta per cell value. For colocalization, the images were processed in the same manner and analysis was performed using the JACoP plugin (Bolte and Cordelieres, 2006) in Fiji, generating Pearson's correlation coefficients, the application of which is indicated in the text and figure legends.

## Live imaging and FRAP

All live imaging and FRAP experiments were performed using the HeLa HA-GLUT4-GFP cell line. Cells were seeded on 35 mm glass bottom dishes (VWR) at standard density, 3–4 days prior to imaging. The day after seeding, the cells were transfected as previously described. On the day of imaging, cells were changed to serum-free media compatible with live imaging: FluoroBrite DMEM (Gibco) for 2 h prior to imaging. This mitigated the effects of insulin contained in serum and enabled measurement of the basal dynamics of the storage compartment. The cells were imaged using the Leica TCS SP8 confocal microscope previously described, with the following environmental modifications: the incubation chamber around the stage was adjusted to a temperature of 37 °C with a 5% $CO_2$ atmosphere. For the FRAP experiments, 8-bit images were taken at a magnification of 0.28 μm pixel−1 ($512 \times 512$ pixels XY-image). In each cell, a single circular region of interest (ROI) of 7.08 μm diameter was used to encompass the area containing the HA-GLUT4-GFP in the perinuclear storage compartment. Photobleaching was performed with 1 scan at 675 nsec pixel−1 using 70% 488 nm laser power resulting in reduction of GLUT4-GFP signal by 70–80%. Images were taken using continuous acquisition at a frame rate of 2 s−1. Prior to bleaching, a sequence of five images was taken, and a total of 100 frames corresponding to 70 s were taken. For each dish, a total of 10–12 cells were photobleached per condition.

## GLUT4 translocation assay using flow cytometry

All translocation experiments used the HeLa HA-GLUT4-GFP (HeLa-G4) cell line. Cells were seeded in 6-well plates at standard density and, the following day, transfected with siRNA for KD and incubated for 72 h thereafter. For the TD rescue experiments, the cells were re-transfected with expression plasmids in fresh media at the 48 h post KD time point and processed 24 h later, at the 72 h endpoint for all FACS experiments. On the day of the experiment, cells were washed (three times, PBS, 37 °C) and serum starved in EBSS (Gibco) for 2 h prior to insulin stimulation. For insulin stimulation, insulin (Sigma) or the same volume of vehicle (water) was added to the wells (20 µl per well, 1% of media volume) to a final concentration of 174 nM, for 20 min at 37 °C. The cells were then placed on ice and washed with PBS containing magnesium and calcium ions (PBS$^{+/+}$) (Corning) which had been pre-cooled to 4 °C, for a total of three times. The cells were then live-stained by blocking for 30 min (PBS$^{+/+}$ with 2% horse serum) on ice and then incubating with anti-HA antibody solution (1:200 diluted in PBS$^{+/+}$ with 2% horse serum) on ice for 1 h to detect surface GLUT4. After this, cells were washed three times in PBS$^{+/+}$ and fixed in 4% PFA for 15 min on ice. After fixation, cells were washed three times in PBS at room temperature. Cells were then incubated with secondary antibody solution (anti-Mouse Alexa Fluor 647 diluted 1:1000 in PBS with 2% horse serum) for 1 h at room temperature followed by washing three times in PBS and gently lifted using a cell scraper (Corning). These suspensions were pelleted by centrifugation at 300×*g* for 3 min and resuspended in a 500 µL of PBS. Flow cytometry data was acquired with FACS Diva acquisition software on a LSRII flow cytometer (Becton Dickinson) equipped with violet (405 nm), blue (488 nm), yellow (561), and red (633 nm) lasers. Typically, 10,000 events were acquired, and median fluorescent intensity (MFI) values for surface GLUT4 (Alexa Fluor 647), CHC22 transfectants (mApple), and total GLUT4 (GFP) were recorded. Post-acquisition analysis was performed using FlowJo software (TreeStar), where debris was removed by using forward/side light scatter gating. Then fluorescence histograms were analyzed, using gating to select only the transfected cell populations in the TD rescue experiments based on the fluorescent signal (mApple) from the expression of the fusion protein. The MFI values were exported and processed in Excel (Microsoft), whereby the ratios of surface-to-total MFI were calculated to quantify the extent of GLUT4 translocation in each condition.

## Protein extraction and immunoblotting

Protein extracts from cells were quantified by BCA (Pierce), separated by SDS-PAGE (10% acrylamide, Merck Millipore), transferred to nitrocellulose membrane (0.2 µm, Bio-Rad), labelled with primary antibodies (1–5 µg/ml), washed, and labelled with species-specific HRP-conjugated secondary antibodies (Thermo Fisher Scientific). HRP activity was detected using Western Lightning Chemiluminescence Reagent (GE Healthcare). The molecular migration position of transferred proteins was compared with the PageRuler Prestain Protein Ladder 10–180 kDa (Thermo Fisher Scientific). Signal quantification was performed using Fiji.

For fluorescent immunoblotting, samples were prepared and processed in the same manner as standard immunoblotting. However, species-specific IRDye-conjugated secondary antibodies were added at a dilution of 1:10,000 (LI-COR Biosciences). These labelled membranes were imaged using the Odyssey DLx imaging system (LI-COR Biosciences). Quantification of the relative abundance of proteins in the membranes was performed using Image Studio (LI-COR Biosciences).

For Coomassie staining, SDS-PAGE gels were stained using InstantBlue Coomassie Protein Stain (Abcam) before imaging using a ChemiDoc MP imaging system (Bio-Rad).

## Immunoprecipitation

Cells from 10 cm, 15 cm, or 500 cm$^2$ dishes were scraped on ice, washed in PBS precooled to 4 °C and pelleted at 300×*g*, for 3 min, at 4 °C. The pellets were resuspended in cold lysis buffer (Table EV1) supplemented with protease and phosphatase inhibitors (2 mM Na$_3$VO$_4$, 1 tab/10 ml; Thermo Fisher Scientific). Cell suspensions were lysed using mechanical shearing by pipetting and then sonicated (10–15 pulses at 50% power). These lysates were centrifuged (16,000×*g* for 15 min, 4 °C) to remove nuclei and cellular debris. The protein content of these lysates was quantified using BCA (Pierce). 1 µg of specific anti-CHC22 antibody (Proteintech) was incubated with 1 ml of 5-10 mg/ml of precleared post-nuclear lysates overnight at 4 °C. The samples were then incubated with washed and pre-blocked protein G Sepharose beads (50 µl, GE Healthcare) for 45 min at 4 °C, before three consecutive washing steps in lysis buffer. The pelleted protein G Sepharose beads were then resuspended in 50 µl of 2× Laemmli sample buffer (Bio-Rad) and proceeded to SDS-PAGE and immunoblotting analysis. Immunoprecipitations were repeated 4–6 times to ensure reproducibility.

## Membrane fractionation

Cells from 10 cm, 15 cm, or 500 cm$^2$ dishes were scraped on ice, washed in PBS precooled to 4 °C and pelleted at 300×*g*, for 3 min, at 4 °C. The pellets were resuspended in fractionation buffer (Table EV1) supplemented with protease inhibitors (1 tab/10 ml; Thermo Fisher Scientific). Cells were mechanically sheared by 20× passages through a 25 G needle (BD Microlance). These lysates were centrifuged (1000×*g* for 10 min, 4 °C) to remove nuclei. The supernatant was then centrifuged (100,000×*g* for 1 h, 4 °C) to separate the cytosolic fraction from the membrane fraction. After this, the supernatant (cytosolic fraction) was removed and both the resulting pellet (Membrane fraction) and a corresponding sample of the cytoplasmic fraction were denatured in either 1× (Membrane fraction) or 4× (Cytoplasmic fraction) Laemmli sample buffer (Bio-Rad) for SDS-PAGE and immunoblotting analysis.

## Protein expression and purification (bacterial)

### CHC fragments
CHC22 Hub (1074-1640) and CHC22 TxD (1521–1640) were cloned into vector backbones derived from pET-Duet and pET15b, respectively, to encode N-terminal 6xHis fusions. CHC22 TD (1-364) and CHC17 TD (1-364) were cloned into a vector backbone derived from pET-Duet as N-terminal His-bdSUMO fusions. CHC22 Hub, CHC22 TD and CHC17 TD were expressed in Rosetta 2 pLysS *E. coli* and induced with 0.4 mM IPTG at 18 °C for 18 h (Hub) or with 0.5 mM IPTG at 37 °C for 4 h (CHC22/17 TD).

CHC22 TxD was expressed in BL21 (DE3) *E. coli* and induced with 0.5 mM IPTG at 30 °C for 5 h.

CHC22 TD (WT and Chim), CHC17 TD and CHC22 TxD cell pellets were lysed by sonication in TD/TxD lysis buffer (Table EV1), and the lysate clarified at 40,000×*g* for 45 min at 4 °C. The fusion proteins were purified using Ni affinity chromatography (HisTrap, Cytiva) in TD/TxD binding buffer before washing with TD/TxD wash buffer and eluting with TD/TxD elution buffer (Table EV1). The eluate was then purified further using size exclusion chromatography (HiLoad 16/30 Superdex 75) in TD/TxD size exclusion buffer (Table EV1). The final product was concentrated to 1–5 mg/ml before supplementation with glycerol to a final concentration of 20%. Proteins were flash-frozen in LN$_2$ and stored at −80 °C until use.

### SNX constructs

Full-length SNX1 (1–522) was sub-cloned from a SNX1 encoding plasmid received as a gift from the Cullen lab (University of Bristol) into a pET15b plasmid backbone as an N-terminal 6xHis fusion. Full-length SNX5 (1–404) was amplified by PCR from human cDNA derived from the GM25256 cell line and cloned into a pGEX3 plasmid backbone as an N-terminal GST-fusion. SNX1 was expressed in BL21 (DE3) *E. coli* and induced with 0.5 mM IPTG at 37 °C for 4 h. SNX5 was expressed in Rosetta 2 pLysS *E. coli* and induced with 0.5 mM IPTG at 18 °C for 18 h.

His-SNX1, GST-SNX5 and GST cell pellets were lysed by sonication in SNX lysis buffer (Table EV1), and the lysate clarified at 40,000×*g* for 45 min at 4 °C. The fusion proteins were purified using affinity chromatography (SNX1; Ni affinity using a HisTrap column Cytiva, SNX5; GST affinity using a GSTrap column, Cytiva) in SNX binding buffer (Table EV1), before washing with either SNX1 wash buffer (SNX1) or SNX binding buffer (SNX5) (Table EV1). SNX1 and SNX5 were subsequently eluted with SNX1 or SNX5 elution buffers (Table EV1), respectively. The eluate was further purified using size exclusion chromatography (Superdex 200 HR 10/300) in SNX size exclusion buffer (Table EV1). The final product was concentrated to 1–5 mg/ml before supplementation with glycerol to a final concentration of 20%. Proteins were flash-frozen in liquid N$_2$ and stored at −80 °C until use.

### Adaptor constructs

Plasmid pIY119 encoding an N-terminal 6xHis p115 fusion was a gift from Igor Yakunin (Munro lab, LMB, Cambridge). p115 (1–962) (amplified from pIY119) and GGA2-hinge-GAE (316–613) were sub-cloned into pGEX3 plasmid backbones as N-terminal GST-fusions. All adaptor constructs (GST, GST-GGA2-HE, GST-p115 and His-p115) were induced in BL21 (DE3) with 0.4 mM IPTG at 18 °C for 18 h. For protein pulldowns, GST-adaptors were affinity-purified onto glutathione beads as described below.

## In vitro binding assay/protein pulldowns

For pulldowns with GST-only or GST-SNX5 (WT) as prey, His-CHC22-Hub, His-CHC22-TxD, and His-p115 were affinity purified using Ni-NTA HisPur beads (Thermo Fisher Scientific) from crude *E. coli* lysate in His purification buffer (Table EV1). For each pulldown, 20 μL of 50% bead slurry was pre-blocked in His PD buffer (Table EV1) for 1 h and incubated with 15 μg of purified GST-only or GST-SNX5, in 250 μL His PD buffer for 45 min. Beads

were then washed three times with 1 ml of His PD wash buffer (Table EV1) and eluted in sample buffer. Pulldowns were analyzed by immunoblotting using antibodies directed against the His-tag, GST, SNX5 and p115. All pulldowns were repeated at least three times to ensure reproducibility.

For competition pulldowns with His-SNX1 and His-CHC22 TxD as prey, GST-SNX5 was affinity purified using glutathione beads (Glutathione Sepharose High Performance, Cytiva) from crude *E. coli* lysate in SNX lysis buffer (Table EV1). The ratio of glutathione beads to GST-SNX5 *E. coli* culture was 50 μl of beads in 2.5 ml of culture. For each pulldown, 5 μl of 50% glutathione beads slurry was pre-blocked in SNX5 blocking buffer (Table EV1). The blocked GST-SNX5 beads were incubated with increasing amounts of purified His-SNX1 in 250 μl of SNX1 PD buffer (Table EV1) for 45 min and subsequently incubated with 2 μg of purified His-CHC22 TxD in 250 μl of TxD PD buffer (Table EV1) for another 45 min. Beads were then washed three times in 1 ml of TxD PD wash buffer (Table EV1) and eluted with sample buffer for analysis. Pulldowns were analyzed by immunoblotting using antibodies directed against the His-tag, CHC22, and GST.

For pulldowns with CHC22 TxD as prey, GST-SNX5 WT and GST-SNX5 S226E were affinity purified using glutathione beads (Glutathione Sepharose High Performance, Cytiva) from crude *E. coli* lysate in SNX binding buffer (Table EV1). For each pulldown, 5 μl of 50% glutathione bead slurry was pre-blocked in SNX5 blocking buffer (Table EV1) and incubated with increasing amounts of purified His-CHC22-TxD, ranging from 4 μg to 16 μg, in 250 μl of TxD PD buffer (Table EV1) for 45 min. Beads were then washed three times with 1 ml TxD PD wash buffer (Table EV1) and eluted with sample buffer for analysis. Pulldowns were analyzed by immunoblotting using antibodies directed against CHC22 and SNX5.

For pulldowns with His-bdSUMO-CHC17 TD or His-bdSUMO-CHC22 TD (WT or Chim) as prey, GST adaptor fusions were affinity-purified using glutathione beads (Glutathione Sepharose High Performance, Cytiva) from crude *E. coli* lysate in GST adaptor buffer (Table EV1). To achieve approximately equal loading of the GST fusion proteins, beads bound to GST proteins were diluted with unbound 'cold' beads (GST; 1:2, GST-GGA2; 2:1, GST-p115; N/A). For pulldowns with CHC22 TD, 10 μl of glutathione of 50% glutathione bead slurry was pre-blocked in 500 μl of CHC22 PD buffer (Table EV1) for 1 h and incubated with 12.5 μg purified His-bdSUMO-CHC22 TD in 250 μl of CHC22 PD buffer (Table EV1) for 45 min. Beads were then washed three times with 1 ml CHC22 PD wash buffer (Table EV1) and eluted with sample buffer for analysis. For pulldowns with CHC17 TD, 20 μl of 50% glutathione bead slurry was pre-blocked in 500 μl CHC17 blocking buffer (Table EV1) for 1 h and incubated with 12.5 μg purified His-bdSUMO-CHC17 TD for 60 min in 250 μl of CHC17 PD buffer (Table EV1). Beads were then washed three times with 1 ml CHC17 PD buffer and eluted with sample buffer for analysis. Pulldowns were analyzed by immunoblotting using antibodies directed against the His-tag and GST.

## Homology modeling (CHC22 TD)

A homology model of the CHC22 TD (1–330) was generated using SWISS-MODEL using a previously solved crystal structure of the CHC17 TD as a template (ter Haar et al, 1998) (PDB: 1BP0).

## AlphaFold modeling

Models of single BAR domains of SNX5 (202–404) and SNX6 (205–406) complexed with a shortened CHC22-Hub (1278–1640) trimer (encoded by three individual chains) were generated using AlphaFold 2-multimer (Version 1.2.0).

## GFP nano-trap precipitations

GFP immunoprecipitations were carried out using GFP-Trap agarose beads (ChromoTek) according to the manufacturer's instructions, with the addition of 10% glycerol into GFP IP lysis and GFP IP dilution buffers. For each GFP immunoprecipitation, a 10 cm² dish was seeded with control HeLa cells at high density (2 million cells/dish) and allowed to attach for 24 h. The following day, cells were transfected with 5 μg plasmid using JetPRIME (Polyplus) according to the manufacturer's instructions. After 18 h, cells were scraped and incubated on ice with gentle agitation for 30 min in 200 μl GFP IP lysis buffer (Table EV1). The lysate was cleared by centrifugation at 17,000 × g for 10 min and was added to 300 μl GFP IP dilution buffer (Table EV1). Diluted lysates were incubated with 20 μl GFP-Trap agarose beads with end-over-end rotation for 1 h at 4 °C. Beads were washed three times and the IPs analyzed by immunoblotting.

## Statistical analysis

Statistical analysis was performed in GraphPad Prism (https://www.graphpad.com/scientific-software/prism/). First, the data were cleaned using ROUT detection of outliers in Prism, followed by testing for normal distribution (D'Agostino - Pearson normality test). Then, the significance for parametric data was tested by either one-way ANOVA followed by Tukey's or Dunnett's post-hoc multiple comparison test, or an unpaired two-tailed $t$ test with Welch's correction. Detailed statistical information for each experiment, including statistical test, number of independent experiments, $P$ values, and definition of error bars are listed in individual figure legends. For GLUT4-GFP FRAP, the bleached ROI, control ROI and background intensity were measured using the 'Time Series Analyzer' plugin in Fiji (NIH) for each time point. These data were then processed in Excel. First, the intensity of the bleached ROI at each time point was background-subtracted and normalized as follows: $In=(Fn-BGn)/(FCn-BGn)$, where: $F_n$ is intensity of the bleached ROI at the time point $n$; $FC_n$, intensity of the control unbleached ROI of the same size at the plasma membrane at the time point $n$; and $BG_n$, background intensity, measured with the same size ROI in cytoplasm at the time point $n$. Then the relative recovery at each time point was calculated using the following formula: $Rn=(In-I1)/(I0-I1)$, where $I_n$, $I_1$ and $I_0$ are normalized intensities of bleached ROI at time point $n$, immediately after photobleaching, and before photobleaching, respectively. These values were input to Prism and nonlinear regression analysis was performed to test for best-fit model and if recoveries were significantly different between cell borders or genotypes. The recovery was fitted to either a single exponential model of the form $f(t)=1-Fim-A1e-t/Tfast$, or a bi-exponential model of the form $f(t)=1-Fim-A1e-t/Tfast-A2e-t/Tslow$, where $F_{im}$ is the size of the immobile fraction, $T_{fast}$ and $T_{slow}$ are the half-times, and $A_1$ and $A_2$ are amplitudes of the fast and slow components of the recovery, respectively. An $F$-test was used to choose the model and compare datasets (Greig and Bulgakova, 2021). Details about graphical representations and error bars are listed in the corresponding figure legends.

## Data availability

The data are available from the corresponding author upon reasonable request.

The source data of this paper are collected in the following database record: biostudies:S-SCDT-10_1038-S44318-024-00198-y.

## Peer review information

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

## Acknowledgements

The authors acknowledge the advice of SM Camus at the initiation of this study and M Giménez-Andrés for coordinating provision of plasmids and reagents, as well as members of the Brodsky lab, WP Bultitude, and L Foltz for intellectual and technical input. We further acknowledge Henrietta Lacks, and the HeLa cell line that was established from her tumor cells, which enabled this work and have made significant contributions to scientific progress and advances in human health. This work was supported by grants to FM Brodsky from the Wellcome Trust (107858/Z/15/Z), UKRI Medical Research Council (MR/S008144/1 and MR/X0183771/1) and UKRI Biotechnology and Biological Sciences Research Council (BB/V001221/1). GT Bates was supported by a Wellcome Trust 4-year interdisciplinary PhD studentship (219856/Z/19/Z). PJ Cullen is supported by the Wellcome Trust (104568/Z/14/Z and 220260/Z/20/Z), the UKRI Medical Research Council (MR/L007363/1 and MR/P018807/1), the Lister Institute of Preventive Medicine, and the award of a Royal Society Noreen Murray Research Professorship (RSRP/R1/211004).

## Author contributions

**Joshua Greig**: Conceptualization; Software; Formal analysis; Supervision; Validation; Investigation; Methodology; Writing—original draft; Writing—review and editing. **George T Bates**: Conceptualization; Software; Formal analysis; Validation; Investigation; Methodology; Writing—original draft; Writing—review and editing. **Daowen I Yin**: Formal analysis; Investigation; Writing—review and editing. **Kit Briant**: Formal analysis; Investigation; Writing—review and editing. **Boris Simonetti**: Conceptualization; Writing—review and editing. **Peter J Cullen**: Conceptualization; Supervision; Funding acquisition; Writing—review and editing. **Frances M Brodsky**: Conceptualization; Resources; Data curation; Supervision; Funding acquisition; Investigation; Methodology; Project administration; Writing—review and editing.

Source data underlying figure panels in this paper may have individual authorship assigned. Where available, figure panel/source data authorship is listed in the following database record: biostudies:S-SCDT-10_1038-S44318-024-00198-y.

## Disclosure and competing interests statement

The authors declare no competing interests.

# Expanded View Figures

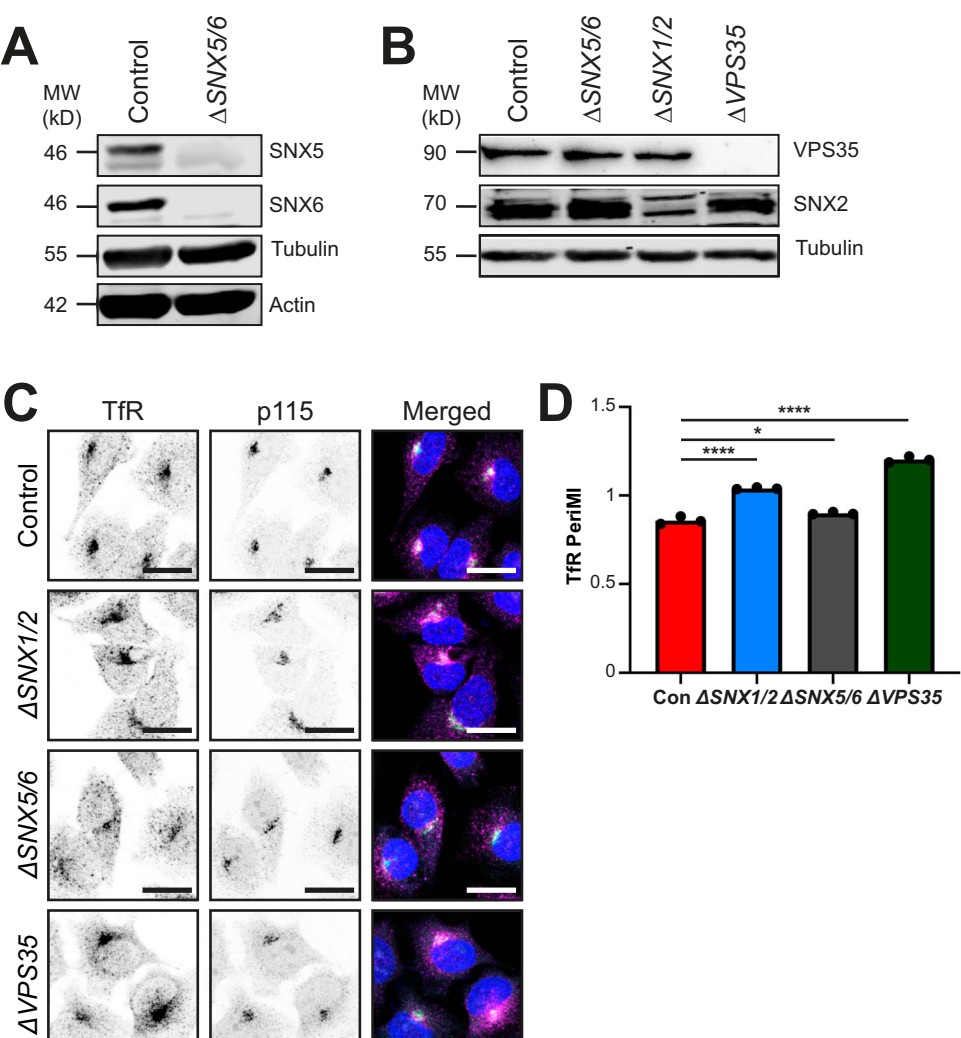

**Figure EV1. Phenotype validation of knock-out HeLa cell lines used in the study and the distribution of transferrin receptor (TfR).**

(A) Representative immunoblot ($n = 2$) of lysates from parental HeLa cells (Control, left) and CRISPR knock-out SNX5/6 HeLa cells (ΔSNX5/6, right) immunoblotted for SNX5, SNX6, tubulin, and actin. The migration positions of MW markers are indicated at the left in kilodaltons (kD). (B) Immunoblot of lysates from parental HeLa (Control, left) and CRISPR knock-out cell lines; SNX5/6, SNX1/2, and VPS35 (ΔSNX5/6, ΔSNX1/2, ΔVPS35) immunoblotted for SNX2, VPS35, and tubulin. The migration positions of MW markers are indicated at the left in kD. (C) Representative images of parental HeLa line (top), ΔSNX1/2 HeLa-derived line (second row), ΔSNX5/6 HeLa-derived line (third row), and ΔVPS35 HeLa-derived line (bottom) immunolabeled for TfR (magenta in merged) and p115 (green in merged) with label overlap in white and DAPI-stained nuclei (blue) in merged. Scale bars: 25 μm. (D) Quantification of PeriMI for TfR in the HeLa lines shown in (A). Each bar represents the mean normalized PeriMI value from three independent experiments, each individually depicted as dots (12–23 cells per genotype, per experiment). Statistical analysis was performed using a one-way ANOVA with a Tukey post-hoc test ($P < 0.0001$(****), $P = 0.0326$ (*)).

## Co-IP HeLa

**Figure EV2. CHC22 is not complexed with the Retromer and ESCPE-1 components VPS35 and SNX1.**

Representative immunoblot ($n = 3$) of CHC22 immunoprecipitate (IP) from HeLa cells. Samples were immunoblotted for CHC22, VPS35, SNX1, and tubulin. Lysate input (5%), bead-only (no antibody) control (Bead) and CHC22 immunoprecipitate (22 IP) are shown. The migration positions of MW markers are indicated at the left in kilodaltons (kD).

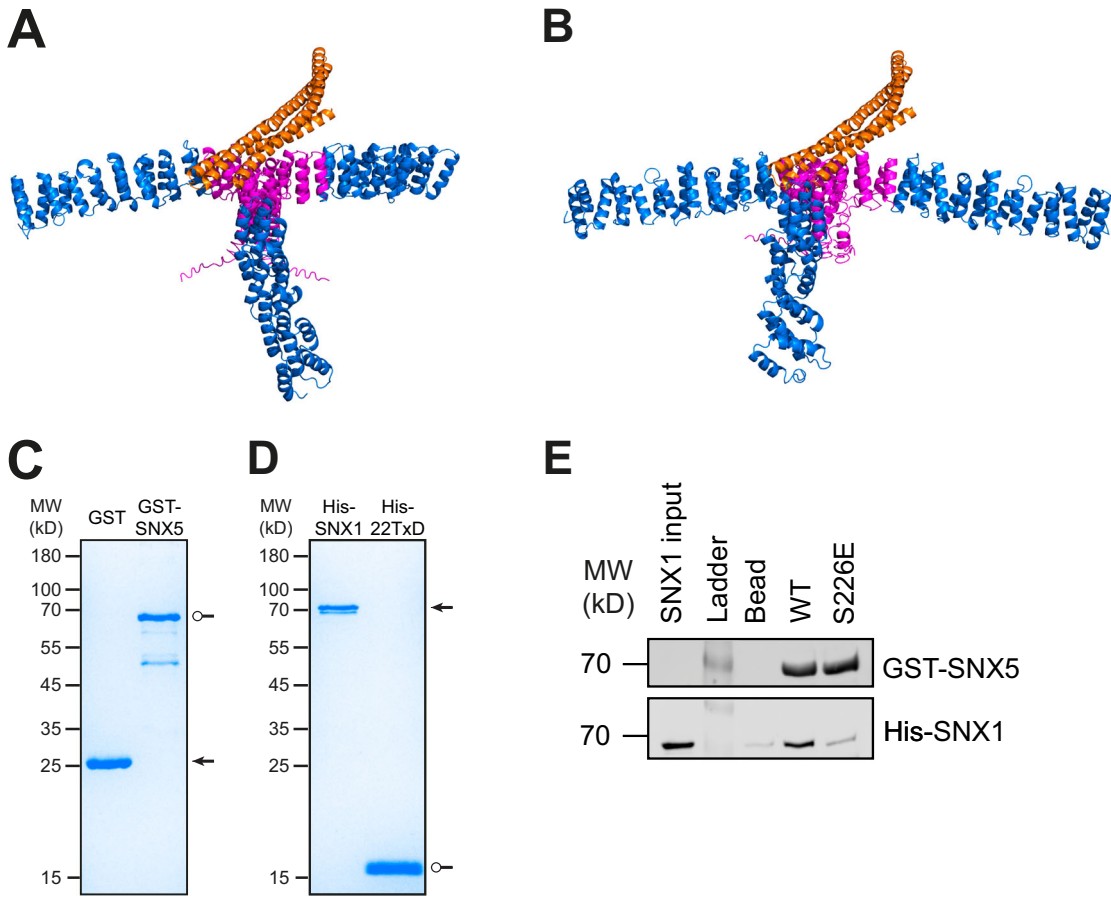

**Figure EV3. Binding of SNX6 to CHC22 TxD predicted by AlphaFold and confirmation of SNX5 mutant loss of binding to SNX1.**

(A, B) Comparison of the AlphaFold-generated models of the interaction between CHC22 Hub (blue proximal leg residues 1278–1519 and magenta TxD residues 1520–1640) and the BAR domain of SNX5 (residues 202–404, orange) (A) or SNX6 (residues 205–406, orange) (B). (C) Coomassie-stained SDS-PAGE gel of GST (arrowhead) and GST-SNX5 (hollow circle) used for in vitro pulldown assays in Figs. 3E,F and 5F. 1 μg protein per lane. (D) Coomassie-stained SDS-PAGE gel of His-SNX1 (arrowhead) and His-22TxD (hollow circle) used in the competition binding assay in Fig. 4D. 1 μg protein per lane. (E) Representative immunoblot (*n* = 2) of the in vitro binding of purified full-length His-SNX1 to immobilized full-length GST-SNX5 wild-type or full-length GST-SNX5 phosphomimetic S226E mutant. His-SNX1 was added to bead-only control (Bead), GST-SNX5 WT (WT), or GST-SNX5 S226E mutant (S226E). Samples were immunoblotted for GST (top) or the His-tag (bottom) and detected proteins indicated at the right. The positions of MW markers (C-E) are indicated in kD at the left.

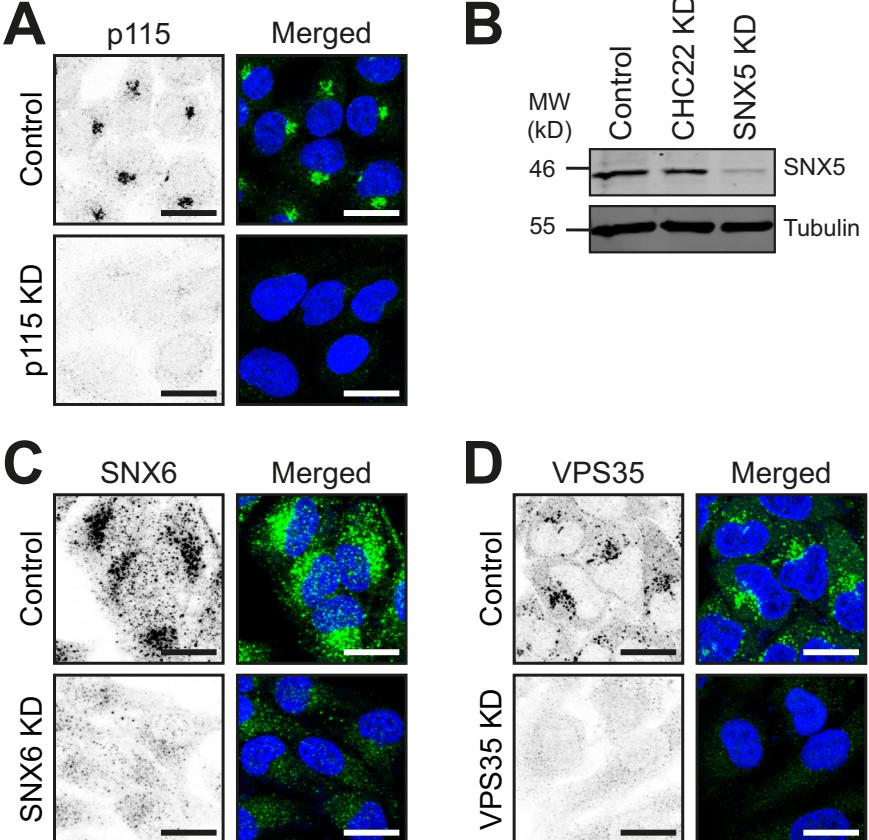

**Figure EV4. Protein depletion by siRNA targeting p115, SNX5, SNX6, and VPS35 used in this study.**

(A) Representative images of HeLa cells transfected with control non-targeting siRNA (Control, top), and cells transfected with siRNA targeting p115 (p115 KD, bottom). Cells were immunolabeled for p115 (green in merged with DAPI-stained nuclei, blue). (B) Representative immunoblot of lysates from HeLa cells transfected with control non-targeting siRNA (Control, left) and HeLa cells transfected with siRNA targeting SNX5 (SNX5 KD, left) immunoblotted for SNX5 and tubulin. The migration positions of MW markers are indicated at the left in kD. (C) Representative images of HeLa cells transfected with control non-targeting siRNA (Control, top) and cells transfected with siRNA targeting SNX6 (SNX6 KD, bottom). Cells were immunolabeled for SNX6 (green in merged with DAPI-stained nuclei, blue. (D) Representative images of HeLa cells transfected with control non-targeting siRNA (Control, top) and cells transfected with siRNA targeting VPS35 (VPS35 KD, bottom). Cells were immunolabeled for VPS35 (green in merged with DAPI-stained nuclei, blue. Scale bars: 25 μm.

