## [Peer Review File · The EMBO Journal]

CHC22 clathrin recruitment to the early secretory pathway requires two-site interaction with SNX5 and p115

Joshua Greig, George Bates, Daowen Yin, Kit Briant, Boris Simonetti, Peter Cullen, and Frances Brodsky

Corresponding author(s): Frances Brodsky (f.brodsky@ucl.ac.uk) , Joshua Greig (josh.greig221b@hotmail.co.uk)

Review Timeline:

Submission Date:	4th Oct 23
Editorial Decision:	13th Oct 23
Appeal Received:	18th Oct 23
Editorial Decision:	14th Dec 23
Revision Received:	4th Jun 24
Editorial Decision:	24th Jun 24
Revision Received:	15th Jul 24
Accepted:	17th Jul 24

Editor: William Teale

Transaction Report:

Dear Frances,

Thank you for submitting your manuscript entitled 'CHC22 clathrin functions in the early secretory pathway by two-site interaction with SNX5 and p115' (EMBOJ-2023-115791) to our editorial office. We have now considered the study within our editorial team, and unfortunately come to the conclusion that we cannot offer publication in The EMBO Journal. Your investigation is a careful dissection of the relationships among SNX5/6, CHC22 and p115 during the ESCPE-1 complex-dependent ER exit. You find that, prior to GSC formation, CHC22 is attached to the ERGIC membrane via a p115 bridge, and these interactions are both direct and via a mutual SNX5/6 interaction. Furthermore, you show that this second interaction precludes SNX5/6 binding to SNX1/2 these data are likely to be of interest to the field. However, while the the analysis that you present provides a fine-grained mechanistic description, we see the level of advance as unlikely to make the study a strong candidate for publication in a broad general journal like The EMBO Journal at this stage.

Given the potential value of your results to researchers directly in the field, we feel that the study should be well-suited for Life Science Alliance (<http://www.life-science-alliance.org/>), our open access journal launched in partnership with Rockefeller University Press and Cold Spring Harbor Laboratory Press. LSA aims to publish solid findings of value to particular communities across all areas of biology. I have therefore briefly discussed the work with the editors of Life Science Alliance, who would indeed be pleased to send your work for in-depth external review, without need for prior reformatting. Should you be interested in this option, please simply follow the link below for transfer. Eric Sawey, Executive Editor of Life Science Alliance (e.sawey@life-science-alliance.org), will be happy to answer any questions you may have.

I am sorry that I cannot be more positive for The EMBO Journal on this occasion, but very much hope that you will find this transfer option worthwhile.

Best wishes,

William

William Teale, PhD
Editor
The EMBO Journal
w.teale@embojournal.org

** As a service to authors, EMBO Press provides authors with the possibility to transfer a manuscript that one journal cannot offer to publish to another EMBO publication or the open access journal Life Science Alliance launched in partnership between EMBO Press, Rockefeller University Press and Cold Spring Harbor Laboratory Press. The full manuscript and if applicable, reviewers' reports, are automatically sent to the receiving journal to allow for fast handling and a prompt decision on your manuscript. For more details of this service, and to transfer your manuscript please click on Link Not Available. **

Dear William,

Thanks for your rapid disposition, which we naturally found very disappointing. I am writing to you in the hope that there is an opportunity to appeal this outcome, as we found it quite surprising. Normally, studies receive criticism for providing insufficient mechanism. In this case, it seems that the advisory board members did not appreciate that in defining the “fine-grained” mechanism of CHC22 recruitment we uncover three novel features that are likely to be of broad interest to several fields. First, this represents a new strategy for cells to achieve Golgi bypass, which is an emerging and important pathway in early secretory trafficking. Second, the repurposing of SNX endosomal coat proteins (which receive a lot of popular coverage) in the early secretory pathway is novel. And third, probably most important, is the unusual dual mechanism we establish for CHC22 recruitment. The dual mechanism we define in this study identifies human-specific targetable sites to address insulin resistance, which is of interest to the glucose metabolism field beyond membrane traffic. For the latter reason alone, we think this work should have broad visibility. In terms of scope and insight, we believe our work is very similar to two recent publications in The EMBO Journal, both of which describe “granular” mechanisms, which have general implications. These are: Yao W, Chen Y, Chen Y, et al. TOR-mediated Ypt1 phosphorylation regulates autophagy initiation complex assembly. EMBO J. 2023;42(19):e112814. doi:10.15252/embj.2022112814<<https://www.embopress.org/doi/full/10.15252/embj.2022112814>>

Das A, Thapa P, Santiago U, et al. A heterotypic assembly mechanism regulates CHIP E3 ligase activity. EMBO J. 2022;41(15):e109566. doi:10.15252/embj.2021109566<<https://www.embopress.org/doi/full/10.15252/embj.2021109566>>

We would like to appeal for the opportunity to have our work reviewed for publication in The EMBO J before being referred to LSA. If questions about impact are still raised, we would welcome further assessment from someone in the metabolism field, as well as someone who could appreciate the contribution to novel mechanisms of membrane traffic and not take the conservative view that this is “just another clathrin paper”.

Thanks for your consideration.

Best wishes,
Frances

Professor Frances M. Brodsky DPhil FMedSci
Research Department of Structural and Molecular Biology
Division of Biosciences
University College London
MRC-LMCB Building, 4th floor, Room 413
Gower Street
London WC1E 6BT
Tel: 44 (0)20 3549 5464 (internal 65464)
Fax:44 (0)20 7679 2682

Dear Frances,

Thank you again for the submission of your manuscript entitled "CHC22 clathrin functions in the early secretory pathway by two-site interaction with SNX5 and p115" (EMBOJ-2023-116791) and for your patience during the review process. We have now received the reports from the referees, which I copy below.

As you can see from their comments, while referee #3 requests some clarification on certain points, all referees are clearly supportive of publication in The EMBO Journal.

I would therefore like to invite you to address the comments of all referees in a revised version of the manuscript. I should add that it is The EMBO Journal policy to allow only a single major round of revision and that it is therefore important to resolve the main concerns at this stage. Please contact me if you have any questions, need further input on the referee comments or if you anticipate any problems in addressing any of their points. Please, follow the instructions below when preparing your manuscript for resubmission.

I would also like to point out that as a matter of policy, competing manuscripts published during this period will not be taken into consideration in our assessment of the novelty presented by your study ("scooping" protection). We have extended this 'scooping protection policy' beyond the usual 3 month revision timeline to cover the period required for a full revision to address the essential experimental issues. Please contact me if you see a paper with related content published elsewhere to discuss the appropriate course of action.

Again, please contact me at any time during revision if you need any help or have further questions.

Thank you very much again for the opportunity to consider your work for publication. I look forward to your revision.

Best regards,

William

William Teale, PhD
Editor
The EMBO Journal
w.teale@embojournal.org

When submitting your revised manuscript, please carefully review the instructions below and include the following items:

- 1) a .docx formatted version of the manuscript text (including legends for main figures, EV figures and tables). Please make sure that the changes are highlighted to be clearly visible.
- 2) individual production quality figure files as .eps, .tif, .jpg (one file per figure).
- 3) a .docx formatted letter INCLUDING the reviewers' reports and your detailed point-by-point response to their comments. As part of the EMBO Press transparent editorial process, the point-by-point response is part of the Review Process File (RPF), which will be published alongside your paper.
- 4) a complete author checklist, which you can download from our author guidelines ([https://wol-prod-cdn.literatumonline.com/pb-assets/embo-site/Author Checklist%20-%20EMBO%20J-1561436015657.xlsx](https://wol-prod-cdn.literatumonline.com/pb-assets/embo-site/Author%20Checklist%20-%20EMBO%20J-1561436015657.xlsx)). Please insert information in the checklist that is also reflected in the manuscript. The completed author checklist will also be part of the RPF.
- 5) Please note that all corresponding authors are required to supply an ORCID ID for their name upon submission of a revised manuscript.
- 6) We require a 'Data Availability' section after the Materials and Methods. Before submitting your revision, primary datasets produced in this study need to be deposited in an appropriate public database, and the accession numbers and database listed under 'Data Availability'. Please remember to provide a reviewer password if the datasets are not yet public (see <https://www.embopress.org/page/journal/14602075/authorguide#datadeposition>). If no data deposition in external databases is needed for this paper, please then state in this section: This study includes no data deposited in external repositories. Note that

the Data Availability Section is restricted to new primary data that are part of this study.

Note - All links should resolve to a page where the data can be accessed.

8) For data quantification: please specify the name of the statistical test used to generate error bars and P values, the number (n) of independent experiments (specify technical or biological replicates) underlying each data point and the test used to calculate p-values in each figure legend. The figure legends should contain a basic description of n, P and the test applied. Graphs must include a description of the bars and the error bars (s.d., s.e.m.).

9) We would also encourage you to include the source data for figure panels that show essential data. Numerical data can be provided as individual .xls or .csv files (including a tab describing the data). For 'blots' or microscopy, uncropped images should be submitted (using a zip archive or a single pdf per main figure if multiple images need to be supplied for one panel). Additional information on source data and instruction on how to label the files are available at .

10) We replaced Supplementary Information with Expanded View (EV) Figures and Tables that are collapsible/expandable online (see examples in <https://www.embopress.org/doi/10.15252/embj.201695874>). A maximum of 5 EV Figures can be typeset. EV Figures should be cited as 'Figure EV1, Figure EV2" etc. in the text and their respective legends should be included in the main text after the legends of regular figures.

12) Our journal encourages inclusion of *data citations in the reference list* to directly cite datasets that were re-used and obtained from public databases. Data citations in the article text are distinct from normal bibliographical citations and should directly link to the database records from which the data can be accessed. In the main text, data citations are formatted as follows: "Data ref: Smith et al, 2001" or "Data ref: NCBI Sequence Read Archive PRJNA342805, 2017". In the Reference list, data citations must be labeled with "[DATASET]". A data reference must provide the database name, accession number/identifiers and a resolvable link to the landing page from which the data can be accessed at the end of the reference. Further instructions are available at .

Further instructions for preparing your revised manuscript:

We realize that it is difficult to revise to a specific deadline. In the interest of protecting the conceptual advance provided by the work, we recommend a revision within 3 months (13th Mar 2024). Please discuss the revision progress ahead of this time with the editor if you require more time to complete the revisions. Use the link below to submit your revision:

Referee #1:

This manuscript presents a detailed and careful characterization of the role of CHC22 clathrin at the ER-Golgi Intermediate compartment (ERGIC). The data show that the sorting nexins SNX5 and SNX6 (which are redundant) bind CHC22 and act in this pathway, which is distinct from the retromer pathway and ESCPE-1 complex. SNX5 also binds directly to p115, and p115 is required to recruit CHC22 to the ERGIC. Intriguingly, the CHC22 N-terminal domain (TD) also binds p115, so that a bipartite interaction is required to recruit CHC22 to ERGIC membranes containing p115. This CHC22-p115 interaction was mapped to a critical patch of residues on CHC22. Finally, both SNX5/6 and CHC22, and particularly the CHC22 TD, are shown to be required for insulin-responsive GLUT4 trafficking in HeLa cells. These results are clearly described, and the discussion section is excellent.

The data define a mechanism that is involved in formation of the insulin-responsive GLUT4-storage compartment at the ERGIC, and further demonstrate how SNX5/6 can function in a complex that is independent of the ESCPE-1 pathway. The work will be important for understanding insulin-regulated GLUT4 trafficking in humans, and also has broader significance for understanding clathrin function and, likely, for targeting of other cargoes from the ERGIC to Golgi-bypass pathways. This is clearly appropriate for the EMBO Journal.

Major concerns: None.

Minor points:

In the HeLa co-IP shown in Fig. 3A, there is some increase in tubulin in the "22 IP" eluate lane, compared to the "bead" control, and the increase in SNX6 is only marginally greater than the difference in tubulin. Figs. 3B and EV2 have very little background, so that together the data are convincing. Nonetheless, if it is possible to obtain a cleaner IP for Fig. 5A, then this might be worthwhile.

Please be sure that the text refers accurately to Fig. 6D and Fig. 6E - in one instance on page 12, Fig 6E is indicated when the description refers to Fig. 6D (this is in the fifth line of the first full paragraph on page 12).

Additional non-essential suggestions:

The work raises some interesting questions. These are beyond the scope of the present manuscript, but may be worth touching upon in the discussion at the authors' discretion. 1) Does the CHC22-SNX5/6-p115 complex play a role only in the formation of the GSC, or is it also involved in ERGIC to cis-Golgi trafficking? Presumably it does not function in ERGIC-Golgi traffic, although this has not been formally demonstrated. 2) If the BAR domain of SNX6 binds to the CHC22 central trimerization domain, rather

than to membrane lipids, then how are membranes remodeled at the ERGIC? Is lipid-binding activity of p115 involved in budding? 3) Because the function of CHC22 is thought to be fulfilled by CHC17 in mice, it would be interesting to know whether mouse CHC17 binds mouse SNX5/6 and p115, and whether these proteins participate in a similar ERGIC-localized complex.

Referee #2:

EMBOJ # (CHC22 clathrin functions in the early secretory pathway by two-site interaction with SNX5 and p115 by Greig et al) provides a multi-faceted analysis of the molecular basis of clathrin isoform, CHC22, localizing to the ER-to-Golgi intermediate compartment (ERGIC). These findings are significant because they provide a molecular basis for the functional difference between two highly similar clathrin isoforms (CHC22 and CHC17).

There is a strong scientific premise for this work, the manuscript is clearly written, the data are of high quality, and the interpretations are supported by the data. There were no detected major weaknesses. Minor weaknesses are listed below. Overall, the enthusiasm for the manuscript is very high.

The authors use quantitative microscopy to demonstrate that SNX5 (and its functionally redundant homology SNX6) are needed for retention of CHC22 to the perinuclear ERGIC. This requirement for SNX5 was shown to be specific for CHC22 as other ERGIC localized proteins are still retained at the ERGIC when SNX5/6 are knocked down. These studies are complemented with biochemical analysis showing that SNX5 binds CHC22, but not the CHC17 isoform.

To better understand the SNX5/CHC22 interaction, the authors look at other SNX5 binding partners - namely SNX1/2. They demonstrate that SNX1/2 and CHC22 compete for binding with SNX5. Additional microscopy experiments demonstrate that SNX5 is the 'molecular bridge' between ERGIC tether p115 and CHC22. Again, a combination of microscopy based assays and biochemical interactions demonstrate the amino terminus of CHC22 is the basis for interacting with p115. Further, the amino terminus of CHC17 does not interact with p115 indicating the point of selectivity. Using these biochemical data, the authors combine amino acid sequence analysis with in silico modeling of the amino termini of CHC22 and CHC17 to predict the amino acids that confer the differences in p115 binding. Substitution of those key amino acids from CHC17 into CHC22 results in a decrease in p115 association, supporting their hypothesis that those amino acids are necessary.

The final set of experiments examine the functional role of CHC22 localizing to the ERGIC. The authors postulate there is a role in generating the insulin-responsive endosomal pool of GLUT4. Using HeLa cells expressing exogenous GFP-tagged GLUT4, the authors use FRAP to demonstrate a role for SNX5/6 in localizing GLUT4 to the perinuclear localization. Further, the authors demonstrate that CHC22 and either SNX5 or SNX6 are required for insulin responsive GLUT4 translocation to the plasma membrane. More specifically, the amino terminus of CHC22 is needed, consistent with this region of CHC22 binding SNX5/6 amino terminal domain.

Minor weaknesses.

The model would be strengthened by testing whether knockdown of CHC17 blocks insulin stimulated GLUT4 translocation.

The methods section states that the GLUT4 translocation assays used water as a vehicle. Are the authors certain that water is the appropriate vehicle and had no effect on the basal GLUT4 translocation? Presumably the insulin was in a buffered solution prior lyophilization. The volume of water added was not indicated and this could affect how much the media was diluted.

Referee #3:

Frances Brodsky and colleagues report in their manuscript that Chc22 clathrin functions in the early secretory pathway by two-site interaction with Snx5 and p115. They arrive at this important conclusion by using a combination of cell biological, biochemical and structural modeling approaches. The results demonstrate that the interaction of Chc22 with Snx5 and p115 at the ERGIC is required to mediate proper formation of GLUT4 storage compartments and GLUT4 trafficking into and out of these GSCs. This work is interesting and I have no major issues that would prevent publication, but a few minor points require clarification prior to publication:

Figure 1H: The data of the subcellular fractionation is not very convincing. The CHC22 re-distribution between membrane and cytosolic fraction is modest at best, and it does not seem to reflect the microscopy experiments. Perhaps the Chc22 protein levels are slightly reduced in the membrane fraction of the Snx5/6 deficient cells, but so are the levels for TfR and Hsc70. This makes it difficult to interpret the data.

Figure 1C / 2A: The localization of Chc22 is obviously changed in Snx5/6 mutant cells and co-localization with p115 is no longer detected in Figure 1C. Yet, in Figure 2A there is still a co-localization of Chc22 with p115 in Snx5/6 mutant cells in a perinuclear

area. Is this a significant variation in the biology of Chc22 localization or just a non-representative image? Also, immunofluorescence staining would suggest that the protein levels of p115 were reduced in snx5/6 ko, which is different in the WB (Figure 1J)?

Figure 2 K: Is there still a signal for SNX6 signal (dots all over the cells) detected in snx5/6 double mutant cells? Is there indeed residual Snx6 protein expressed? If so, how much compared to control cells?

Figure 4D - E: Please clarify the relative protein stoichiometries used for the competition experiments. It will facilitate the interpretation of the results. Also, to further test the model, it would be important to perform similar competition experiments with Snx5 mutants that no longer bind to Snx1. Also, would such a competition model work in vivo (is there a rough idea about Snx1 Snx5 Chc22 molecule (ball park figures) numbers?).

Figure 6 A/D: The microscopy experiments in figure 6A show that Chc22delTD no longer co-localizes with p115. Yet the two proteins still robustly interact by co-IP experiments (Figure 6D). Please explain the discrepancy.

Figure 8 F. The protein levels of GLUT4 appear to be dramatically reduced in cells expressing Chc22delTD? Is GLUT4 destabilized and degraded?

page 9: bracket (is missing. generated models that were near identical to that of SNX5 binding to CHC22 Fig. EV3A, B).

Point by point responses (*bold italics*) to Reviewers' Comments (plain font) with specific figure changes noted:

Reviewer #1:

This manuscript presents a detailed and careful characterization of the role of CHC22 clathrin at the ER-Golgi Intermediate compartment (ERGIC). The data show that the sorting nexins SNX5 and SNX6 (which are redundant) bind CHC22 and act in this pathway, which is distinct from the retromer pathway and ESCPE-1 complex. SNX5 also binds directly to p115, and p115 is required to recruit CHC22 to the ERGIC. Intriguingly, the CHC22 N-terminal domain (TD) also binds p115, so that a bipartite interaction is required to recruit CHC22 to ERGIC membranes containing p115. This CHC22-p115 interaction was mapped to a critical patch of residues on CHC22. Finally, both SNX5/6 and CHC22, and particularly the CHC22 TD, are shown to be required for insulin-responsive GLUT4 trafficking in HeLa cells. These results are clearly described, and the discussion section is excellent.

The data define a mechanism that is involved in formation of the insulin-responsive GLUT4-storage compartment at the ERGIC, and further demonstrate how SNX5/6 can function in a complex that is independent of the ESCPE-1 pathway. The work will be important for understanding insulin-regulated GLUT4 trafficking in humans, and also has broader significance for understanding clathrin function and, likely, for targeting of other cargoes from the ERGIC to Golgi-bypass pathways. This is clearly appropriate for the EMBO Journal.

Major concerns: None.

Minor points:

In the HeLa co-IP shown in Fig. 3A, there is some increase in tubulin in the "22 IP" eluate lane, compared to the "bead" control, and the increase in SNX6 is only marginally greater than the difference in tubulin. Figs. 3B and EV2 have very little background, so that together the data are convincing. Nonetheless, if it is possible to obtain a cleaner IP for Fig. 3A, then this might be worthwhile.

We appreciate the reviewer's point about the slight increase in tubulin background in HeLa cells, which can occasionally vary. Like the reviewer, our confidence in the specificity for SNX5 and SNX6 is further based on the cleaner co-IPs from myotubes. Unfortunately, we were unable to obtain a "cleaner" example from HeLa cells, having exhausted our directly labelled secondary reagent that we conjugated ourselves that was used to detect SNX6 by blotting. And we prefer to use the current data, given that the same reagent was used to analyse both HeLa and myotubes at the same time, and results shown were reproduced at least three times.

Please be sure that the text refers accurately to Fig. 6D and Fig. 6E - in one instance on page 12, Fig 6E is indicated when the description refers to Fig. 6D (this is in the fifth line of the first full paragraph on page 12).

We have corrected this error and double checked the other figure panel references in the text of the revised manuscript.

Additional non-essential suggestions:

The work raises some interesting questions. These are beyond the scope of the present manuscript, but may be worth touching upon in the discussion at the authors' discretion.

- 1) Does the CHC22-SNX5/6-p115 complex play a role only in the formation of the GSC, or is it also involved in ERGIC to cis-Golgi trafficking? Presumably it does not function in ERGIC-Golgi traffic, although this has not been formally demonstrated.

As noted in the Discussion, we have not formally excluded a broader function of this complex in other Golgi-bypass pathways. However, we think that this complex is probably not involved in ERGIC to cis-Golgi traffic. Our previous work demonstrated that CHC22 trafficking of GLUT4 is insensitive to Brefeldin A (BFA) and using the RUSH assay we detected no trafficking of GLUT4 to the cis-Golgi marker GM130 after release (Camus et al., 2020). Thus, our investigations of CHC22 cargo or pathogens that use the pathway suggest that it is a Golgi-bypass pathway and not involved in the conventional secretory route.

- 2) If the BAR domain of SNX6 binds to the CHC22 central trimerization domain, rather than to membrane lipids, then how are membranes remodeled at the ERGIC? Is lipid-binding activity of p115 involved in budding?

As noted in the Discussion, it appears that the CHC22-SNX interaction is incompatible with a membrane remodelling function for the BAR domain. The reviewer suggests that there must be another mechanism for membrane remodelling. Membrane curvature could, of course, be induced by the CHC22 coat itself, as observed for the mechanics of CHC17. However, we do not exclude the possibility that there are other membrane-remodelling elements recruited by CHC22, and it's an interesting suggestion that p115 itself could be involved. While establishing how CHC22 induces membrane remodelling is beyond the scope of the current work, it is certainly of interest for future studies.

- 3) Because the function of CHC22 is thought to be fulfilled by CHC17 in mice, it would be interesting to know whether mouse CHC17 binds mouse SNX5/6 and p115, and whether these proteins participate in a similar ERGIC-localized complex.

We agree entirely with the reviewer and have preliminary results that, in rodents, CHC17 splicing may enable this interaction, so we are actively pursuing this question!

Reviewer #2:

EMBOJ # (CHC22 clathrin functions in the early secretory pathway by two-site interaction with SNX5 and p115 by Greig et al) provides a multi-faceted analysis of the molecular basis of clathrin isoform, CHC22, localizing to the ER-to-Golgi intermediate compartment (ERGIC). These findings are significant because they provide a molecular basis for the functional difference between two highly similar clathrin isoforms (CHC22 and CHC17).

There is a strong scientific premise for this work, the manuscript is clearly written, the data are of high quality, and the interpretations are supported by the data. There were no detected major weaknesses. Minor weaknesses are listed below. Overall, the enthusiasm for the manuscript is very high.

The authors use quantitative microscopy to demonstrate that SNX5 (and its functionally redundant homology SNX6) are needed for retention of CHC22 to the perinuclear ERGIC. This requirement for SNX5 was shown to be specific for CHC22 as other ERGIC localized proteins are still retained at the ERGIC when SNX5/6 are knocked down. These studies are complemented with biochemical analysis showing that SNX5 binds CHC22, but not the CHC17 isoform.

To better understand the SNX5/CHC22 interaction, the authors look at other SNX5 binding

partners – namely SNX1/2. They demonstrate that SNX1/2 and CHC22 compete for binding with SNX5. Additional microscopy experiments demonstrate that SNX5 is the ‘molecular bridge’ between ERGIC tether p115 and CHC22. Again, a combination of microscopy based assays and biochemical interactions demonstrate the amino terminus of CHC22 is the basis for interacting with p115. Further, the amino terminus of CHC17 does not interact with p115 indicating the point of selectivity. Using these biochemical data, the authors combine amino acid sequence analysis with in silico 3D modelling of the amino termini of CHC22 and CHC17 to predict the amino acids that confer the differences in p115 binding. Substitution of those key amino acids from CHC17 into CHC22 results in a decrease in p115 association, supporting their hypothesis that those amino acids are necessary.

The final set of experiments examine the functional role of CHC22 localizing to the ERGIC. The authors postulate there is a role in generating the insulin-responsive endosomal pool of GLUT4. Using HeLa cells expressing exogenous GFP-tagged GLUT4, the authors use FRAP to demonstrate a role for SNX5/6 in localizing GLUT4 to the perinuclear localization. Further, the authors demonstrate that CHC22 and either SNX5 or SNX6 are required for insulin responsive GLUT4 translocation to the plasma membrane. More specifically, the amino terminus of CHC22 is needed, consistent with this region of CHC22 binding SNX5/6 amino terminal domain.

Minor weaknesses.

The model would be strengthened by testing whether knockdown of CHC17 blocks Insulin stimulated GLUT4 translocation.

We agree with the reviewer that this is an important point. We performed this experiment in three previously published studies (Vassilopoulos et al., 2009; Camus et al., 2020; Fumagalli et al., 2019). In the latter two, we showed that CHC17 knock-down does not impair insulin induced translocation of GLUT4 in the same HeLa GLUT4-GFP model system used in this current study. In the first study, this was addressed in differentiated myoblasts using radioactive glucose uptake. In the revised manuscript, we have now included a sentence referencing these previous studies, to strengthen this point about the functional differences between CHC17 and CHC22.

The methods section states that the GLUT4 translocation assays used water as a vehicle. Are the authors certain that water is the appropriate vehicle and had no effect on the basal GLUT4 translocation? Presumably the insulin was in a buffered solution prior lyophilization. The volume of _{water} added was not indicated and this could affect how much the media was diluted.

We have now included details about the volume of water added to the methods section of the revised manuscript. The lyophilized insulin was reconstituted in water to produce a 100X stock. 20µL of this stock or water were added to 2mL of media per well of cells. Thus, the amount of water or insulin added was 1% of the assay volume, so would have a limited effect on the osmolarity.

Reviewer #3:

Frances Brodsky and colleagues report in their manuscript that Chc22 clathrin functions in the early secretory pathway by two-site interaction with Snx5 and p115. They arrive at this important conclusion by using a combination of cell biological, biochemical and structural modeling approaches. The results demonstrate that the interaction of Chc22 with Snx5 and p115 at the ERGIC is required to mediate proper formation of GLUT4 storage compartments and GLUT4 trafficking into and out of these GSCs. This work is interesting and I have no major issues that would prevent publication, but a few minor points require clarification prior

to publication:

Figure 1H: The data of the subcellular fractionation is not very convincing. The CHC22 re-distribution between membrane and cytosolic fraction is modest at best, and it does not seem to reflect the microscopy experiments. Perhaps the Chc22 protein levels are slightly reduced in the membrane fraction of the Snx5/6 deficient cells, but so are the levels for TfR and Hsc70. This makes it difficult to interpret the data.

In this experiment, we measured the ratio of membrane to cytosolic protein for each protein assessed, which is independent of total levels of the protein in cells. To make this clear, we have changed the presentation of the data. Instead of reporting M/C ratios, we now show the percentage of total protein in the membrane fraction (Membrane/Membrane + Cytosol for each protein for each cell type. With this quantification, it is more obvious that the membrane levels of CHC22 are reduced in the SNX5/6 null cells compared to the WT cell. In addition, we have now included quantification of the Hsc70 control marker, which is partially cytosolic and membrane-associated. Like CHC17 and TfR, the membrane distribution of Hsc70 is not significantly different between the two cell types. We further note that inhibition of CHC22 association with early secretory membranes is not expected to affect CHC22 association with endosomal membranes through different adaptor interactions reported in previous studies (Esk et al., 2010; Camus et al., 2020). Therefore, we would expect only partial loss of membrane association when interfering with the CHC22-p115 complex, as the endosomal association would still be present. Figure edit: Figure 1G is replotted data now including Hsc70 (formerly Figure 1I).

Figure 1C / 2A: The localization of Chc22 is obviously changed in Snx5/6 mutant cells and co-localization with p115 is no longer detected in Figure 1C. Yet, in Figure 2A there is still a co-localization of Chc22 with p115 in Snx5/6 mutant cells in a perinuclear area. Is this a significant variation in the biology of Chc22 localization or just a non-representative image? Also, immunofluorescence staining would suggest that the protein levels of p115 were reduced in snx5/6 ko, which is different in the WB (Figure 1J)?

We appreciate these comments, and they inspired us to further evaluate our immunofluorescence data sets, as well as the co-localization with p115.

Regarding the quantitative analysis of the PeriMI distribution of CHC22, we found that our analysis methodology was not the most appropriate for this form of data. To improve this, we have now re-evaluated the same data such that the PeriMI for proteins of interest is calculated and expressed as an average PeriMI for all cells (10-29 from 4 different fields) per genotype per experiment (n=3), rather than a single cell-by cell basis from all 3 experiments as in the original analysis, so that the statistical analysis is now correct. Importantly, the re-calculation did not change the results and confirmed a significant reduction in CHC22 PeriMI in the SNX5/6 null cells and an increase in the SNX1/2 null cells as well as in the rescue experiments with expression of SNX5 after transfection.

Figure edits for revised statistical analysis:

- ***Figure 1 Figure 1C (formerly 1D), 1E (formerly 1G), 1I (formerly 1I), 1K (formerly 1M); Removed part of former Figure 1A, Figures 1B, B', 1E as not relevant to revised data analysis, so original panels have new labels***
- ***Figure 2B, 2C, 2D, 2G, 2H, 2I, 2L***
- ***Figure 5D, 5E***
- ***Figure EV1D (formerly EV1B)***
- ***Figure 6C, 6E (changed to bar format, no revision of analysis)***

Regarding effects on p115, we also noted that p115 is less concentrated in the PeriMI area in the SNX5/6 null cells, which given what we establish with the biochemical experiments in this study, may be due to a loss of CHC22 coats. This dispersal would account for the apparent reduction in p115 staining, but still be consistent with the blot showing no change in protein levels.

Regarding the images compared in Panels 1B (formerly 1C) and 2A, there is some slight variation in CHC22 staining levels on an experiment-by-experiment basis, as noted by the reviewer. We further note that the immunofluorescence imaging is simply the starting point for the follow up biochemical studies that revealed individual molecular interactions consistent with the images obtained, and that imaging has the limitation of visualising multiple CHC22 pathways at once.

Figure edit:

Figure 1B: The image in Figure 1B (formerly 1C) and 1J (formerly 1L) were replaced to be more compatible with staining in Figure 2A.

Figures 1H (formerly 1J) contrast adjusted to correlate with new panels above.

Figure 2 K: Is there still a signal for SNX6 signal (dots all over the cells) detected in snx5/6 double mutant cells? Is there indeed residual Snx6 protein expressed? If so, how much compared to control cells?

The SNX5/6 (Null) cells were generated by CRISPR engineering and previously characterised by the Cullen lab (Simonetti et al, 2017). These cells were also phenotyped in the Brodsky lab by immunoblotting and the absence of SNX5/6 protein was confirmed. We now include this confirmatory data in Figure EV1A of the revised manuscript and include analysis of the other knockout cell lines used in Figure EV1B. There is some background signal with the anti-SNX6 antibody in the SNX5/6 nulls cells (Fig. 2K) with a mean value that is 8% that of the control cells (Fig. 2L). This is typical of background noise and the pattern does not resemble the SNX6 distribution observed in the control cells, so we do not think this represents residual protein. Figure addition: Figure EV1A, B.

Figure 4D - E: Please clarify the relative protein stoichiometries used for the competition experiments. It will facilitate the interpretation of the results. Also, to further test the model, it would be important to perform similar competition experiments with Snx5 mutants that no longer bind to Snx1. Also, would such a competition model work in vivo (is there a rough idea about Snx1 Snx5 Chc22 molecule (ball park figures) numbers?).

In the competition assay, SNX5 is purified onto glutathione beads from lysate, so the precise amount bound is hard to accurately determine. However, defined amounts of TxD and SNX1 were added so the relative protein stoichiometries of these can be calculated. A fixed 2 μ g of TxD was added to SNX5 after pre-incubation with 1, 2, 4 and 8 μ g of SNX1 – representing molar ratios of approximately 8, 4, 2, 1 :1 (TxD : SNX1). Saturation of SNX5 with SNX1 should represent 1:1 binding according to studies of the ESCPE-1 complex.

The reviewer's suggestion to do further studies with the SNX5 mutant (S226E) that does not bind SNX1 was extremely useful. In our initial study, looking at a single high concentration of this mutant, we found that it bound CHC22 TxD similarly to the WT SNX5. However, when we did titrations of protein in preparation for the suggested inhibition experiment, we found that TxD binding to the SNX5 (S226E) mutant was not

as strong as binding to WT SNX5. Thus, the mutation that affects SNX1 binding (confirmed in new Fig. EV3E) also partially affects CHC22 binding. This further establishes that SNX1 and CHC22 interact with SNX5 at the same interface, but with some different SNX5 residues involved. Instead of the suggested control inhibition experiment, we have now included this updated finding in the revised version of the manuscript as Figure 4E & F, showing the relative interaction of CHC22 with the WT and mutant SNX5. The Results section has been modified accordingly. While the SNX5 mutant is a phosphomimetic, this mutation can affect both partners and there is no reliable knowledge about endogenous protein levels, and how they might compete and regulation by phosphorylation is less likely, as mentioned in the Discussion. Figure edits:

- **Figure 4E and F (replacing original 4E)**
- **Figure EV3E added to show mutant loss of binding to SNX1**

Figure 6 A/D: The microscopy experiments in figure 6A show that Chc22delTD no longer co-localizes with p115. Yet the two proteins still robustly interaction by co-IP experiments (Figure 6D). Please explain the discrepancy.

This query emphasises the nature of the dual binding mechanism described in this study. The Delta TD CHC22 showed significantly reduced co-localization and Co-IP with p115, but some interaction of CHC22 with p115 remains due to the secondary interaction of CHC22 with p115 via SNX5/6.

Figure 8 F. The protein levels of GLUT4 appear to be dramatically reduced in cells expressing Chc22delTD? Is GLUT4 destabilized and degraded?

We have previously published data showing that, in the absence of CHC22, GLUT4 is not directed towards the storage compartment and is partially degraded (Vassilopoulos et al., 2009). The data cited by the reviewer is consistent with this previous observation as Delta TD CHC22 is not functional for rescuing formation of the GLUT4 storage compartment, so some degradation of GLUT4 would be expected.

page 9: bracket (is missing. generated models that were near identical to that of SNX5 binding to CHC22 Fig. EV3A, B).

We have corrected this error in the revised manuscript.

Dear Frances,

Thank you submitting a revised version of your manuscript. It was sent to the same three reviewers that originally appraised your work; their comments are attached to the bottom of this email. As you will see, all three referees are satisfied with the changes you made. Before we can move forwards towards publication of your manuscript, though, there are some remaining editorial points which need to be addressed. In this regard, would you please:

change the title of the 'Conflict of Interests' statement to the 'Disclosure and Competing Interests Statement',
remove the author credit section from the manuscript
either upload Table EV5, or ammend callout to it,
address the apparent slice-and-shifts in Figures 3E and 5F (this is not seen in the source data) and reconvert files if necessary,
provide source data for tubulin loading controls shown in EV1B and EV4B as these images appear to be identical,
provide exact p values in the legends of figures 1c, e, g, i, k; 2b-d, g-i, l; 5d-e; 6c; 8b, e, g; EV 1d,
correct the mismatch between the annotated p values in the figure legend and the annotated p values for figure EV 1d,
describe the nature of entity for 'n' in the legends of figures 1g; 5f; 6e, and
define black arrows in the legend of figure EV 3c-d.

I look forward to receiving these changes. EMBO Press is an editorially independent publishing platform for the development of EMBO scientific publications.

Best wishes,

William

William Teale, PhD
Editor
The EMBO Journal
w.teale@embojournal.org

We realize that it is difficult to revise to a specific deadline. In the interest of protecting the conceptual advance provided by the work, we recommend a revision within 3 months (22nd Sep 2024). Please discuss the revision progress ahead of this time with the editor if you require more time to complete the revisions. Use the link below to submit your revision:

Referee #1:

The authors have done a very nice job of addressing previous critiques and, in my opinion, the manuscript is ready for publication.

Referee #2:

The original submission of EMBOJ-2023-115791 was viewed favorably by all three reviewers. The minor recommendations were adequately addressed by the authors. The biggest changes including re-ordering and re-analyzing some figures for a more accurate statistical analysis. None of the findings of the manuscript changed.

Minor Point:

Figure 2 has part of a graph at the bottom of the page that looks to be leftover from the original submission.

Referee #3:

The authors have done excellent work during the revision, and addressed my points.

Point by point responses (Blue Text) to editorial changes requested (Black Text).

1. Change the title of the 'Conflict of Interests' statement to the 'Disclosure and Competing Interests Statement',

We have changed this title in the revised (R2) manuscript.

2. Remove the author credit section from the manuscript

This has been removed from the revised (R2) manuscript

3. Either upload Table EV5, or amend callout to it,

We have uploaded a Word (.doc) format for table EV5 to ensure that it is editable, in accordance with the EMBO Journal author guidelines.

4. Address the apparent slice-and-shifts in Figures 3E and 5F (this is not seen in the source data) and reconvert files if necessary.

We were puzzled by how this band distortion occurred when preparing the main figures – as you noted, this distortion is not present in the original source images. We have remade the main panel figures using a “making a mask” function rather than cropping in Adobe Illustrator and that appears to have addressed the distortion problem with these bands. We include these remade panels in the revised Figures 3 and 5 (R2 versions).

5. Provide source data for tubulin loading controls shown in EV1B and EV4B as these images appear to be identical

An error occurred with the labelling of the blots for figure EV1B and EV4B. There was a mislabelling between a SNX5 Knock-Out (KO) and Knock-down (KD). Consequently, the blot in EV4B showed the SNX5 KO cells rather than the correct SNX5 KD. We have corrected this by replacing EV4B with the correct blot in the revised figure submission (Fig. EV4 R2) and for consistency remade EV1B with the same source data (Fig. EV1 R2). Furthermore, we have prepared and include the source data files for the immunoblot panels in EV1B and EV4B. We apologise for this mistake on our part and thank the editorial team for bringing it to our attention.

6. Provide exact p values in the legends of figures 1c, e, g, i, k; 2b-d, g-i, l; 5d-e; 6c; 8b, e, g; EV 1d,

We have added the exact p values to the corresponding figure legends of the revised (R2) manuscript.

7. Correct the mismatch between the annotated p values in the figure legend and the annotated p values for figure EV 1d,

We have edited the legend of Figure EV1 to include the p value star indicators for that Figure alone, rather than all the star value brackets which are not relevant directly to Figure EV1.

8. Describe the nature of entity for 'n' in the legends of figures 1g; 5f; 6e, and

The n number in these cases refers to separate biological replicates, we have included this definition in the revised figure legends.

Define black arrows in the legend of figure EV 3c-d.

These arrows indicated the proteins of interest, but this was not explicit in the R1 version. We have included additional details in the revised (R2) Figure legend for Figure EV3. Moreover, we have further distinguished the indicated bands by making one of the arrows for each gel a hollow circle head.

Comment from Referee 2:

Figure 2 has part of a graph at the bottom of the page that looks to be leftover from the original submission.

This residual graph image has been removed.

Dear Frances and Joshua,

I am pleased to inform you that your manuscript has been accepted for publication in the EMBO Journal.

Congratulations! I will be really glad to see this work in The EMBO Journal.

Best wishes,

William

William Teale, PhD
Editor
The EMBO Journal
w.teale@embojournal.org
